# Continual Multimodal Contrastive Learning

**Xiaohao Liu**    **Xiaobo Xia**$^*$    **See-Kiong Ng**    **Tat-Seng Chua**
National University of Singapore
xiaohao.liu@u.nus.edu    {xbx, seekiong, dcscts}@nus.edu.sg

## Abstract

Multimodal Contrastive Learning (MCL) advances in aligning different modalities and generating multimodal representations in a joint space. By leveraging contrastive learning across diverse modalities, large-scale multimodal data enhances representational quality. However, a critical yet often overlooked challenge remains: multimodal data is rarely collected in a single process, and training from scratch is computationally expensive. Instead, emergent multimodal data can be used to optimize existing models gradually, *i.e.*, models are trained on a sequence of modality pair data. We define this problem as Continual Multimodal Contrastive Learning (CMCL), an underexplored yet crucial research direction at the intersection of multimodal and continual learning. In this paper, we formulate CMCL through two specialized principles of stability and plasticity. We theoretically derive a novel optimization-based method, which projects updated gradients from dual sides onto subspaces where any gradient is prevented from interfering with the previously learned knowledge. Two upper bounds provide theoretical insights on both stability and plasticity in our solution. Beyond our theoretical contributions, we conduct experiments on multiple datasets by comparing our method against advanced continual learning baselines. The empirical results further support our claims and demonstrate the efficacy of our method. Our codes are available at https://github.com/Xiaohao-Liu/CMCL.

## 1   Introduction

Humans perceive the world in a multimodal way. They capture and process information from what they see, hear, smell, and touch. Machines are evolving towards multisensory processing to mimic this capability by leveraging specialized latent representations to interpret the world effectively [1, 2, 3, 4, 5]. Multimodal contrastive learning (MCL) builds upon unimodal [6, 7, 8] and cross-modal contrastive learning [9, 10], which demonstrates exceptional representational power and broad applicability. The core principle behind it is to bring modality pairs (*e.g.*, vision-text or vision-audio data) from the same instance closer together while simultaneously pushing apart representations of different instances. Consequently, MCL constructs a unified representation space yet capable of accommodating diverse modalities [11, 12, 13, 14].

Extensive data support is critical for ensuring representational quality. Recent studies have focused on either compensating to the existing modality data [13, 14] or introducing the emergent ones [15]. Besides learning from scratch, collected data can also be incrementally introduced to gradually enhance a model's capabilities, thereby resolving two significant challenges: 1) the difficulty of collecting all prepared multimodal data in a single process, and 2) the high computational expense associated with training on mixed datasets from initialization. With the emergent multimodal data, multimodal models can be progressively elevated via continual/incremental learning. Unfortunately, recent research in MCL primarily advances in an "*all in one go*" manner. Moreover, it is intractable

---

$^*$Corresponding author.

39th Conference on Neural Information Processing Systems (NeurIPS 2025).

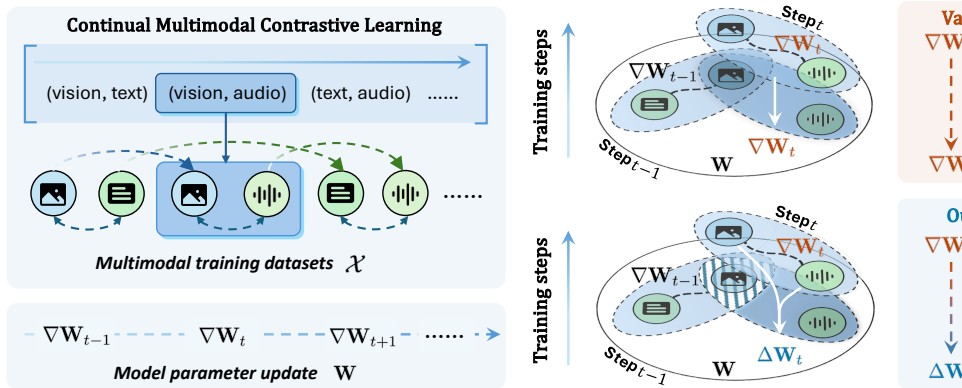

Figure 1: **The problem of Continual Multimodal Contrastive Learning (CMCL).** Multimodal models learn from data step by step. At each step, the two modalities are aligned through contrastive learning. Different pairs of modalities are involved at different steps. The model parameters are continually updated through gradients from different training steps.

Figure 2: **Our gradient-projection method (bottom) compared with the vanilla (top).** Within the parameter space spanned by $\mathbf{W}$, the gradient updated at the previous step $\nabla\mathbf{W}_{t-1}$ is directly overridden in vanilla training. Instead, our method projects the new gradient $\nabla\mathbf{W}_t$ onto the non-effective subspaces of the old, denoting $\Delta\mathbf{W}_t$, to maintain both stability and plasticity.

for current continual learning methods to address this problem [16, 17, 18, 19], due to the inter-modal complexity and the task-agnostic constraints, where the objective of contrasting different modalities remains consistent over time[2].

Aware of this, we pioneer the formulation of continual multimodal contrastive learning (CMCL) in this paper. In CMCL, a sequence of multimodal data is utilized for training. At each training step, two modalities are aligned through contrastive learning, and the new modality pair is subsequently integrated into the following step, as illustrated in Figure 1. We specify two definitions of *stability* and *plasticity* in CMCL. Here stability refers to the model's capability to retain acquired knowledge over previously interacted modalities, while plasticity denotes that the model can learn effectively from new modality pairs. Building upon this foundation, we theoretically derive a novel CMCL method by projecting updated parameter gradients from dual sides onto specialized subspaces, as shown in Figure 2. We project gradients based on both their own modality knowledge and their interacted ones, *i.e.*, a dual-sided projection. These inter-modality histories are utilized to construct the corresponding gradient projectors. Incorporating new data at the current training step within these projected subspaces does not adversely impact model performance on datasets from previous steps. Furthermore, we provide theoretical guarantees by establishing two bounds addressing stability and plasticity. These bounds substantiate our method's effectiveness theoretically, further supported by empirical analyses. Besides, we showcase the flexibility of our method by extending it to any steps and any modality pairs thus achieving both efficacy and efficiency in practice. We conduct experiments on 7 datasets to evaluate our solution for classification and retrieval tasks. The results well justify our claims. The proposed method consistently achieves superior performance in plasticity and stability. Compared with previous state-of-the-art (SOTA) baselines of the continual learning field, it demonstrates remarkable results across multiple metrics.

Before delving into details, we clearly emphasize our contributions as follows:

- Conceptually, we take the initiative to formulate continual multimodal contrastive learning explicitly and provide specialized definitions of stability and plasticity that guide methodological development. Our work establishes a foundational framework for exploring this previously unaddressed problem and encourages future contributions to CMCL.

- Technically, we introduce a novel method that operates on the parameter gradients. These gradients are projected onto the subspaces where any updated gradients have no effect on the previous knowledge. This method is built upon existing modality-binding models and can be readily extended to any training steps and modality pairs in the practical training procedure.

---

[2]We review the literature related to this work in Appendix D.

- Theoretically, we rigorously derive our method based on the fundamental requirements (stability and plasticity), supported by detailed theoretical proofs. Additionally, we provide two theoretical bounds that guarantee the objectives, which can be tightened in practice to reinforce our claim.

- Empirically, we conduct extensive experiments on 7 datasets to evaluate the performance in the CMCL setting. By utilizing three modality-binding methods as backbones, we demonstrate superior performance compared to SOTA continual learning baselines, which confirms the effectiveness of our solution.

## 2 Preliminaries

**Multimodal learning.** Typical multimodal learning can be formulated as:

$$\mathbb{P}(\mathbf{x}, y) \triangleq \mathbb{P}_{y|\mathbf{x}}\left(y \mid h \circ g(\mathbf{x})\right) \mathbb{P}_{\mathbf{x}}(\mathbf{x}), \tag{1}$$

where $g$ is the multimodal encoder that generates corresponding representation $\mathbf{z}$ given the input $\mathbf{x}$, and $h$ is the task mapping (*e.g.*, linear classifier [10, 11] or even large language models (LLMs) [20, 21, 22]). In general, $g$ is modal-agnostic. Here we use $g_m$ as the modal-specific encoder, indicating that we do not make any assumption about a shared model architecture or parameters. For different modalities, a unified representation space is desired to facilitate the development of $h$, so that one well-trained $h$ serves all $\{g_m\}_{m \in \mathcal{M}}$.

**Modality-binding (modality pairs for training).** Note that $g$ is normally implemented by modality-binding methods to obtain representations for different modalities in a unified space. These methods utilize paired data $\mathcal{X}^{m;m'} = \{(\mathbf{x}_1^m, \mathbf{x}_1^{m'}), (\mathbf{x}_2^m, \mathbf{x}_2^{m'}), \dots\} \subset \mathcal{X}^m \times \mathcal{X}^{m'}$ to optimize the contrastive objective derived from CLIP [10]:

$$\mathcal{L} = -\log \frac{\exp(\mathbf{z}_i^{m\top} \mathbf{z}_i^{m'}/\tau)}{\exp(\mathbf{z}_i^{m\top} \mathbf{z}_i^{m'}/\tau) + \sum_{j \neq i} \exp(\mathbf{z}_i^{m\top} \mathbf{z}_j^{m'}/\tau)}, \tag{2}$$

where $\tau$ is the temperature hyperparameter. The two modalities $m$ and $m'$ from one instance $i$ are pulled closer while being pushed away from different instances $j$. Despite having only two modalities in each batch, more diverse datasets are introduced to achieve multimodal contrastive learning with more paired modality data. The typical utilization of modality-binding is exemplified by CLIP, which is contrastive learning for bi-modality. Here only two modalities (*i.e.*, image/vision and text) are used for training. However, several works [11, 15, 13] have recently extended this approach with more modalities on different modality pair data, *e.g.*, image-audio or audio-text pairs.

**Range-Null space.** Given a matrix $\mathbf{A}$, its pseudo-inverse $\mathbf{A}^\dagger$ satisfies $\mathbf{A}\mathbf{A}^\dagger\mathbf{A} \equiv \mathbf{A}$. $\mathbf{A}^\dagger\mathbf{A}$ projects $\mathbf{b}$ to the *range space* of $\mathbf{A}$, formally, $\mathbf{A}(\mathbf{A}^\dagger\mathbf{A})\mathbf{b} \equiv \mathbf{A}\mathbf{b}$. $(\mathbf{I} - \mathbf{A}^\dagger\mathbf{A})$ can be seen as the *null space* projector that directly satisfies $\mathbf{A}(\mathbf{I} - \mathbf{A}^\dagger\mathbf{A})\mathbf{b} \equiv \mathbf{0}$. And there are several ways to resolve the pseudo-inverse $\mathbf{A}^\dagger$, for instance, in singular value decomposition (SVD), $\mathbf{A}^\dagger = \mathbf{V}\mathbf{\Lambda}^\dagger\mathbf{U}^\top$ for $\{\mathbf{U}, \mathbf{\Lambda}, \mathbf{V}^\top\} = \mathrm{SVD}(\mathbf{A})$, where $\mathbf{\Lambda}^\dagger$ take the reciprocal of the nonzero singular values in $\mathbf{\Lambda}$ while leaving zeros unchanged. Null space projection is used in prior works like Adam-NSCL [16, 23, 24], which constructs the null space projector based on layer-wise features. Despite the inspiration, it is not capable of tackling the inter-modality complexity in CMCL. A specialized solution is still desirable for CMCL.

## 3 Continual Multimodal Contrastive Learning

### 3.1 Problem Setup and Two Goals

**Problem setup.** Suppose we have a sequence of datasets $\mathcal{X} = \{\mathcal{X}_1^{m;m'}, \mathcal{X}_2^{m;m'}, \dots\}$, where $\mathcal{X}_1^{m;m'}$ denotes the dataset containing modalities $m$ and $m'$ at the initial step and the sequence length is $N$. The parameters of each modality encoder $g_m$ are optimized with different pairs, signifying $\mathbf{W}_0^m, \dots, \mathbf{W}_N^m$. Note that $\mathbf{W}_i^m = \mathbf{W}_{i+1}^m$ when $\mathcal{X}_{i+1}$ does not include the modality $m$. Without the loss of generality, we consider two continuous training datasets, denoted as $\mathcal{X}_{t-1}^{m_1,m_2}$ and $\mathcal{X}_t^{m_1,m_3}$, which share at least one common modality. The corresponding encoder $g_{m_1}$ will be continually

optimized with different data. Here we formulate multimodal learning on multiple datasets in a sequential learning paradigm:

$$\mathbf{W}_t^{m_1} = \arg\min_{\mathbf{W}_{t-1}^{m_1}} \sum_{(\mathbf{x}_t^{m_1}, \mathbf{x}_t^{m_3}) \in \mathcal{X}_t^{m_1; m_3}} \mathcal{L}(g_{m_1}(\mathbf{x}_t^{m_1}; \mathbf{W}_{t-1}^{m_1}), g_{m_3}(\mathbf{x}_t^{m_3}; \mathbf{W}_{t-1}^{m_3})). \tag{3}$$

Note that $\mathbf{W}_{t-1}^{m_1}$ is trained on $\mathcal{X}_{t-1}^{m_1, m_2}$. Besides, if $m_3 := m_2$, this formulation can be generalized to the case in CLIP [10] and the bi-modality continual training [25, 26, 27].

**Two goals in CMCL.** The main goal is to maintain alignment stability along with continual learning. This alignment is typically defined as the inner product between different modalities of the same instance [10, 11, 15], signifying $\mathbf{z}_i^{m\top}\mathbf{z}_i^{m'}$. This formulation is also utilized in multimodal contrastive learning as mentioned in Eq. (2).

**Definition 1** (Stability). *Given two adjacent training datasets $\mathcal{X}_{t-1}^{m_1, m_2}$ and $\mathcal{X}_t^{m_1, m_3}$, the model satisfies stability if:*

$$\underbrace{(\mathbf{Z}_{t-1;t}^{m_1})^\top \mathbf{Z}_{t-1;t}^{m_2}}_{model\ at\ current\ step\ t} = \underbrace{(\mathbf{Z}_{t-1;t-1}^{m_1})^\top \mathbf{Z}_{t-1;t-1}^{m_2}}_{model\ at\ previous\ step\ t-1}, \tag{4}$$

*where $\mathbf{Z}_{t-1;t}^{m_1}$ denotes the representations for modality $m_1$ in $(t-1)$-th dataset generated by the model trained after step $t$. $\mathbf{Z}_{t-1;t-1}^{m_1}$ represents the representations generated by the model immediately trained on the dataset at its own step $t-1$.*

For the dataset used previously (*i.e.*, $\mathcal{X}_{t-1}^{m_1, m_2}$), the current model parameterized by $\mathbf{W}_t$ can perform well aligned with the previous model parameterized by $\mathbf{W}_{t-1}$. For simplicity, we denote the alignment scores between two modalities (*i.e.*, $m_1$ and $m_2$) as $\mathbf{A}_{t-1;t}^{m_1, m_2}$, utilizing the model at step $t$ for dataset $\mathcal{X}_{t-1}$. Consequently, we simplify Eq. (4) to $\mathbf{A}_{t-1;t}^{m_1, m_2} = \mathbf{A}_{t-1;t-1}^{m_1, m_2}$.

**Definition 2** (Plasticity). *The model has the capability to learn from new datasets, adhering to the objective of multimodal contrastive learning: $\mathbf{Z}_{t;t}^{m\top}\mathbf{Z}_{t;t}^{m'} \sim p_t^{m, m'}$, where $p_t^{m, m'}$ is the distribution that samples the $t$-th dataset for two different modalities.*

Another goal indicates that the model, even with constraints to memorize the old, can still have the plasticity to accept the new. Specifically, given the dataset at the current step $t$, the parameters of the model can still be optimized by a multimodal contrastive learning objective to fit its distribution.

**Dilemmas.** We consider the post-training paradigm to elevate the existing pre-trained models that are already trained with various datasets. Current methods without a continual learning objective face a significant challenge of *forgetting*: the alignment plasticity is broken when the model is trained with new datasets (*i.e.*, $\mathbf{A}_{t-1;t}^{m_1, m_2} \neq \mathbf{A}_{t-1;t-1}^{m_1, m_2}$).

## 3.2 Dual-sided Null Space (DNS) for CMCL

In this section, we propose projecting gradients to the dual-sided null space to satisfy both stability and plasticity, as depicted in Figure 3.

**Stability requirement.** Recall the stability requirement in CMCL of $(\mathbf{Z}_{t-1;t}^{m_1})^\top \mathbf{Z}_{t-1;t}^{m_2} = \mathbf{A}_{t-1;t-1}^{m_1, m_2}$. That is to say, given the same modality pair data from the previous step, the model trained on new datasets should maintain the alignments on modality pairs from earlier steps. This implies that new updates (*i.e.*, gradients) should not interfere with effective parts within the parameter space.

**Plasticity requirement.** According to the contrastive learning objective in Eq. (3), unconstrained gradients naturally encourage parameters optimized from previous datasets to adapt

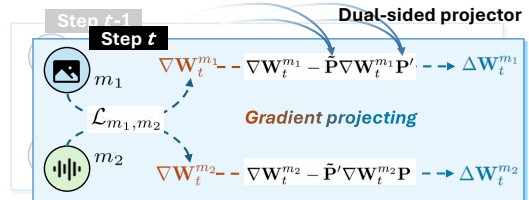

Figure 3: **The paradigm of the proposed method (DNS) for CMCL.** At the current step $t$, gradients ($\nabla \mathbf{W}_t^{m_1}$ and $\nabla \mathbf{W}_t^{m_2}$) derived by contrastive learning between modalities $m_1$ and $m_2$ are projected from dual sides to $\Delta \mathbf{W}_t^{m_1}$ and $\Delta \mathbf{W}_t^{m_2}$. The projectors are built upon the features from the previous steps to constrain the gradients. By substituting for the gradients, model parameters are optimized from new modality pair data while retraining the prior knowledge.

to new datasets. Therefore, plasticity emerges directly from the inherent learning process. In vanilla CMCL, where the model sequentially trains on datasets without any explicit constraints, plasticity is inherently maintained. Introducing additional regularization strategies inherently increases risks to plasticity.

**Theoretical motivation.** Building upon the above analyses, we explore gradient updates in detail to establish theoretical motivations. Let us consider the left-hand side of Eq. (4), *i.e.*, $(\mathbf{Z}_{t-1;t}^{m_1})^\top \mathbf{Z}_{t-1;t}^{m_2}$. We introduce the prior encoders of $g_m^*$ (*cf.*, plasticity in Definition 2) and a linear operator that transfers the priors $\mathbf{Z}_{t-1;*}^{m_1}$ and $\mathbf{Z}_{t-1;*}^{m_2}$, with $\mathbf{Z}_{t-1;t}^{m_1} = \mathbf{W}_t^{m_1}\mathbf{Z}_{t-1;*}^{m_1}$ and $\mathbf{W}_t^{m_1} \in \mathbb{R}^{d\times d}$, $\mathbf{Z}_{t-1;*}^{m_1} \in \mathbb{R}^{d\times n}$, where $d$ is the feature dimension and $n$ denotes the batch size. Besides, we consider the gradient update rule as $\mathbf{W}_t^m = \mathbf{W}_{t-1}^m - \eta\nabla\mathbf{W}_t^m$, where $\eta$ is the learning rate. Therefore, we have:

$$
\begin{aligned}
& (\mathbf{Z}_{t-1;t}^{m_1})^\top \mathbf{Z}_{t-1;t}^{m_2} \\
& = (\mathbf{Z}_{t-1;*}^{m_1})^\top \underbrace{(\mathbf{W}_t^{m_1})^\top \mathbf{W}_t^{m_2}}_{\text{global parameters at step } t} \mathbf{Z}_{t-1;*}^{m_2} = (\mathbf{Z}_{t-1;*}^{m_1})^\top (\mathbf{W}_{t-1}^{m_1} - \eta\nabla\mathbf{W}_t^{m_1})^\top (\mathbf{W}_{t-1}^{m_2} - \eta\nabla\mathbf{W}_t^{m_2})\mathbf{Z}_{t-1;*}^{m_2} \\
& = (\mathbf{Z}_{t-1;*}^{m_1})^\top \underbrace{(\mathbf{W}_{t-1}^{m_1})^\top \mathbf{W}_{t-1}^{m_2}}_{\text{global parameters at step } t-1} \mathbf{Z}_{t-1;*}^{m_2} \\
& \quad - \eta(\mathbf{Z}_{t-1;*}^{m_1})^\top \underbrace{\left((\mathbf{W}_{t-1}^{m_1})^\top \nabla\mathbf{W}_t^{m_2} + (\nabla\mathbf{W}_t^{m_1})^\top \mathbf{W}_{t-1}^{m_2} - \eta(\nabla\mathbf{W}_t^{m_1})^\top \nabla\mathbf{W}_t^{m_2}\right)}_{\text{updated global parameters } \tilde{\mathbf{W}}} \mathbf{Z}_{t-1;*}^{m_2}.
\end{aligned} \tag{5}
$$

**Projecting $\tilde{\mathbf{W}}$ (globally).** We consider the parameter update globally, where $\tilde{\mathbf{W}}$ is used to update $(\mathbf{W}_{t-1}^{m_1})^\top \mathbf{W}_{t-1}^{m_2}$ via $(\mathbf{W}_t^{m_1})^\top \mathbf{W}_t^{m_2} = (\mathbf{W}_{t-1}^{m_1})^\top \mathbf{W}_{t-1}^{m_2} - \eta\tilde{\mathbf{W}}$. To keep *stability* for the previous datasets, we need to find a new parameter update $\bar{\mathbf{W}}$ that satisfies $(\mathbf{Z}_{t-1;*}^{m_1})^\top \bar{\mathbf{W}}\mathbf{Z}_{t-1;*}^{m_2} = \mathbf{0}$. Therefore, $\mathbf{A}_{t-1;t}^{m_1,m_2} = \mathbf{A}_{t-1;t-1}^{m_1,m_2} + \mathbf{0}$.

**Theorem 3.** *Let the global parameter update follow* $(\mathbf{W}_t^{m_1})^\top \mathbf{W}_t^{m_2} = (\mathbf{W}_{t-1}^{m_1})^\top \mathbf{W}_{t-1}^{m_2} - \eta\bar{\mathbf{W}}$, *and the original parameter update be denoted as:*

$$
\tilde{\mathbf{W}} = \left((\mathbf{W}_{t-1}^{m_1})^\top \nabla\mathbf{W}_t^{m_2} + (\nabla\mathbf{W}_t^{m_1})^\top \mathbf{W}_{t-1}^{m_2} - \eta(\nabla\mathbf{W}_t^{m_1})^\top \nabla\mathbf{W}_t^{m_2}\right). \tag{6}
$$

*Then, the stability holds if* $\bar{\mathbf{W}} := \tilde{\mathbf{W}} - \mathbf{P}'\tilde{\mathbf{W}}\mathbf{P}$*, where* $\mathbf{P}'$ *and* $\mathbf{P}$ *are the space projectors for* $\mathbf{Z}_{t-1;*}^{m_1}$ *and* $\mathbf{Z}_{t-1;*}^{m_2}$*, respectively. See proof in Appendix B.1.*

**Remark.** Once obtaining the global gradient update $\tilde{\mathbf{W}}$, we are able to find the optimal solution to project it such that $(\mathbf{Z}_{t-1}^{m_1})^\top \bar{\mathbf{W}}\mathbf{Z}_{t-1}^{m_2} = \mathbf{0}$. $\bar{\mathbf{W}}$ is the closest point to $\tilde{\mathbf{W}}$ but lies in the null space. In other words, the projected global gradient does not cause any effect on previous modality pairs at step $t - 1$. Here we call the space spanned by $\tilde{\mathbf{W}} - \mathbf{P}'\tilde{\mathbf{W}}\mathbf{P}$ as null space as well[a].

**Discussion on $\bar{\mathbf{W}}$.** $\bar{\mathbf{W}}$ is derived from the original $\tilde{\mathbf{W}}$ while removing its part relevant to the previous training. Such relevance is probed and extracted via pre-curated projectors corresponding to the dual-side modality data $\mathbf{Z}_{t-1}^{m_1}$ and $\mathbf{Z}_{t-1}^{m_2}$ (see Section 2 Range-Null space for constructing projectors). This projection reflects the modality interaction in multimodal contrastive learning, where the parameters are optimized from both modalities corresponding to different sides of the parameter matrix $\tilde{\mathbf{W}}$ (*i.e.*, the row space for $m_1$ and column space for $m_2$).

---

[a]The formal definition of null space is the solution space for $\mathbf{A}\mathbf{x} = \mathbf{0}$, differing from our problem.

**Projecting $\nabla\mathbf{W}$ (locally).** Following this, we hope to update the local gradients in practice, *i.e.*, optimizing the model parameters corresponding to each modality, to satisfy the stability. Here we find the deviation of these gradients, thus approaching the $\tilde{\mathbf{W}} - \mathbf{P}'\tilde{\mathbf{W}}\mathbf{P}$ lies in the null space.

**Theorem 4.** *Let the local parameter update follow* $\mathbf{W}_t^m = \mathbf{W}_{t-1}^m - \eta\Delta\mathbf{W}_t^m$*, the stability holds if*

$$
\begin{cases}
\Delta\mathbf{W}_t^{m_1} := \nabla\mathbf{W}_t^{m_1} - \tilde{\mathbf{P}}\nabla\mathbf{W}_t^{m_1}\mathbf{P}', \\
\Delta\mathbf{W}_t^{m_2} := \nabla\mathbf{W}_t^{m_2} - \tilde{\mathbf{P}}'\nabla\mathbf{W}_t^{m_2}\mathbf{P},
\end{cases} \tag{7}
$$

*where* $\tilde{\mathbf{P}}'$ *and* $\tilde{\mathbf{P}}$ *are the space projectors for* $\mathbf{Z}_{t-1;t-1}^{m_1}$ *and* $\mathbf{Z}_{t-1;t-1}^{m_2}$*, respectively. See proof in Appendix B.2.*

**Remark.** We find the local gradient transformation for $\nabla \mathbf{W}_t^m, \forall m \in \{m_1, m_2\}$ to achieve the global gradient update $\tilde{\mathbf{W}} - \mathbf{P}'\tilde{\mathbf{W}}\mathbf{P}$. In this case, we can transform the gradient from independent models (*i.e.*, different encoders), yet still keep the stability.

**Discussion on $\Delta \mathbf{W}_t^m$.** We introduce an additional projector of the previously generated features, denoted by $\mathbf{Z}_{t-1;t-1}^{m'}$. $\Delta \mathbf{W}_t^m$ is similarly derived from the original gradient $\nabla \mathbf{W}_t^m$ while removing the parts determined by both the own input modality data $\mathbf{Z}_{t-1;*}^m$ (*cf.*, Theorem 3) and the generated features of another modality $\mathbf{Z}_{t-1;t-1}^{m'}$. $\tilde{\mathbf{P}}$ and $\tilde{\mathbf{P}}'$ project vectors onto modal-specific *output* subspaces, which reside within a unified representation space due to the contrastive learning paradigm. Despite sharing a unified space, the subspaces of different modality features are not exactly identical. Therefore, we employ two separate projectors for a more precise projection. This projection reflects the effective mapping from the modality input to the aligned output space.

### 3.3 The Theoretical Guarantee: Bounds and Capabilities

After inferring the method to achieve CMCL, we examine the performance guarantees theoretically. Besides, we introduce the extensions for any steps and any modality pairs during the practical training procedure. Note that we also provide empirical support for our theoretical results in Section 4.2.

**Theorem 5** (The upper bound of loss of stability). *In CMCL, the upper bound of loss of stability is:*

$$\|\mathbf{A}_{t-1;t}^{m_1,m_2} - \mathbf{A}_{t-1;t-1}^{m_1,m_2}\|_2 \leq \eta^2 \cdot \|\mathbf{Z}_{t-1;*}^{m_1}\|_2 \|\mathbf{Z}_{t-1;*}^{m_2}\|_2 \cdot \mathcal{F}(\nabla \mathbf{W}), \tag{8}$$

*where $\mathcal{F}$ represents the interactions between different updated gradients $\nabla \mathbf{W}_t^{m_1}$ and $\nabla \mathbf{W}_t^{m_2}$. Specifically, $\mathcal{F}(\nabla \mathbf{W}) := 2\|\nabla \mathbf{W}_t^{m_1}\|_2 \|\nabla \mathbf{W}_t^{m_2}\|_2 + \|\nabla \mathbf{W}_t^{m_1}\|_2^2 + \|\nabla \mathbf{W}_t^{m_2}\|_2^2$. See proof in Appendix B.3.*

**Remark.** The loss of stability is primarily bounded by the square of the learning rate $\eta^2$, and the norm of the input features $\{\mathbf{Z}_{t-1;*}^m\}$. The input features are typically normalized to stabilize the prior encoders' training. Therefore, their norm does not become so large as to interfere with the bound. Furthermore, low-dimensional or sparse features have been observed to better preserve basic objectives for instance discrimination [28], potentially tightening the bound. A smaller learning rate results in a tighter bound as well. This is reasonable when considering the extreme case ($\eta = 0$), where the parameters are not updated during training without any loss of stability.

**Theorem 6** (The upper bound of plasticity). *By transforming the gradients, the loss $\mathcal{L}_t$ update at step $t$ from the previous loss $\mathcal{L}_{t-1}$ is following:*

$$\mathcal{L}_t - \mathcal{L}_{t-1} \leq 0 \tag{9}$$

*to keep the plasticity when $\frac{o(\eta)}{\eta} \leq 0$ at the time $t$. See proof in Appendix B.4.*

**Remark.** The loss decreases, indicating learning from the new datasets. $\mathcal{L}_t = \mathcal{L}_{t-1}$ if and only if the row space of $\tilde{\mathbf{W}}$ lies in the column space of $\mathbf{Z}_{t-1}^{m_1}$ and the column space of $\tilde{\mathbf{W}}$ lies in the column space of $\mathbf{Z}_{t-1}^{m_2}$. In this case, $\tilde{\mathbf{W}}$ can only make effects to the features *within* the space of previous datasets. However, $\tilde{\mathbf{W}}$ is calculated via current datasets $\mathbf{Z}_t^m$, which makes the equality rare.

**Extension for any steps $\{\mathbf{A}_{i;*}^{m,m'}\}_{<t}$.** In general, on continual learning, we hope to maintain stability for any seen datasets. Therefore, we have:

$$\mathbf{A}_{i;t}^{m_1,m_2} = \mathbf{A}_{i;i}^{m_1,m_2}, \quad \forall i < t. \tag{10}$$

That means we need to project $\tilde{\mathbf{W}}$ onto the null space of all previous datasets, via $\mathbf{P}_{<t}$. We utilize the uncentered feature covariance [16] to obtain the null space projector via SVD approximation (Section 2). Let $\bar{\mathbf{Z}}_{<t}^m = \frac{\bar{n}_{t-2}}{\bar{n}_{t-1}}\bar{\mathbf{Z}}_{<t-1}^m + \frac{n_{t-1}}{\bar{n}_{t-1}}\tilde{\mathbf{Z}}_{t-1}^m$, where $\tilde{\mathbf{Z}}_{t-1}^m \triangleq \frac{1}{n_{t-1}}(\mathbf{Z}_{t-1}^m)^\top \mathbf{Z}_{t-1}^m$ (the equality is proved in Appendix B.5). Similarly, for global gradient updates, $\bar{\mathbf{W}} := \tilde{\mathbf{W}} - \mathbf{P}'_{<t}\tilde{\mathbf{W}}\mathbf{P}_{<t}$ for extension. For the local gradient update, we are required to maintain an additional feature covariance constructed from the previous models. We can achieve this in efficiency. For instance, $\Delta \mathbf{W}_t^{m_2} := \nabla \mathbf{W}_t^{m_2} - \tilde{\mathbf{P}}'_{<t}\nabla \mathbf{W}_t^{m_2}\mathbf{P}_{<t}$, where $\tilde{\mathbf{P}}'_{<t}$ is the space projector for $\{\mathbf{Z}_{i;i}^{m_1}\}_{i<t}$. Note that, the guarantee for stability and plasticity can be easily extended to any step, as we only modify the projectors that are able to project the features onto the space that covers all previous datasets.

Table 1: The statistics of datasets for modality pair training, which involve modalities of V↔Vision, A↔Audio, T↔Text, VI↔VIdeo, TH↔THermal, TA↔TActile, and D↔Depth.

| Dataset | Modalities | | | | | | | Evaluation | Tasks | | Modes | #Examples | #Classes |
|---|---|---|---|---|---|---|---|---|---|---|---|---|---|
| | V | A | T | VI | TH | TA | D | | Retrieval | Classification | | | |
| UCF101 | - | - | ✓ | ✓ | - | - | - | Acc | - | ✓ | Train, Test | 8,080 | 101 |
| ESC50 | - | ✓ | ✓ | - | - | - | - | Recall, Acc | ✓ | ✓ | Train, Test | 2,000 | 50 |
| NYUDv2 | ✓ | - | - | - | - | - | ✓ | Recall | ✓ | - | Train, Test | 48,238 | - |
| VGGSound-S | - | ✓ | ✓ | ✓ | - | - | - | Recall, Acc | ✓ | ✓ | Train, Test | 12,000 | 309 |
| Clotho | - | ✓ | ✓ | - | - | - | - | Recall | ✓ | - | Train, Test | 4,885 | - |
| TVL | ✓ | - | ✓ | - | - | ✓ | - | Recall | ✓ | - | Train, Test | 43,741 | - |
| LLVIP | ✓ | - | - | - | ✓ | - | - | Recall | ✓ | - | Train, Test | 15,488 | - |

**Extension for any modality pairs** $\{m, m'\}_{<t}$**.** In particular, on CMCL, diverse modality pairs are involved to optimize corresponding encoders. In this case, we can simply extend the previous gradient update on $\{m_1, m_2\}$ in step $t-1$ and $t$ to any different adjacent modality pairs. Suppose training on modality pairs of $\{m_1, m_2\}$ at step $t-1$ and $\{m_1, m_3\}$ at step $t$. If $m_3 := m_2$, it follows Theorem 4. If $m_3 \neq m_2$, we set $\nabla \mathbf{W}_t^{m_2} = \mathbf{0}$. Furthermore, the corresponding feature projector update can be omitted as well to save computational cost. Therefore, only features at relevant steps are required to be updated rather than considering all steps. The algorithm flow and pseudo-code of training and inference procedures of our method are provided in Appendix A.1.

# 4 Experiments

## 4.1 Experimental Setups

**Tasks & datasets.** We evaluate on 7 multimodal datasets, including UCF101 [29], ESC50 [30], NYUDv2 [31], VGGSound-S [32][3], Clotho [33], TVL [34], and LLVIP [35]. For performance comparison, we categorize the evaluation tasks as retrieval and classification. The important statistics are provided in Table 1. More details are provided in Appendix A.2. For datasets with more than two modalities, we split them into bi-modal ones to cater to the modality pair training. For instance, VGGSound-S will be reorganized into three versions (VI-A, VI-T, and T-A). As a result, 11 training steps are involved in our training and evaluation.

**Implementation details.** We utilize pre-trained modality-binding methods as priors, followed by linear learners for each modality for training. Specifically, we employ ImageBind [11], Language-Bind [15], and UniBind [13], which are already trained with a sequence of modality pairs and are capable of generating multimodal representations in a joint space. Upon these powerful priors, we construct the projection module that projects the novel knowledge onto the dual-sided null space of the previous one within the trainable parameters. We utilize PyTorch [36] for implementation. The model is optimized via an AdamW optimizer [37] with a learning rate of 0.0001 and weight decay of 0.001. The batch size is set to 64. We approximate the null space projection with truncated SVD, where a minimum eigenvalue $\lambda_{\min}$ is set to 0.01 for ImageBind and Unibind, while 0.0001 for LanguageBind. See more implementation details in Appendix A.3, A.5, and E.

**Baselines.** We consider various methods from continual learning fields. Specifically, we select *Vanilla* (continual training without any strategy) [11], Gradient Episodic Memory (*GEM*) [38], Dark Experience Replay (*DER & DER++*) [39], Elastic Weight Consolidation (*EWC*) [40], Contrastive Continual Learning (*Co²L*) [17], *C-Flat* [41], and Contrastive Incremental Learning with Adaptive distillation (*CILA*) [18]. Notably, our method is replay-free, which maintains efficiency. The feature covariances are updated only at the end of each training step. We adapt these methods for the CMCL setting. More details of the baselines are provided in Appendix A.4 and E.

**Evaluation metrics.** We consider the evaluation metrics from both plasticity (learnability) and stability (anti-forgetting) perspectives. For plasticity, we employ *Recall@k* for cross-modal retrieval tasks, where $k$ is set in the range of $\{1, 5, 10\}$. Besides, we use average accuracy (*Acc*) for classification tasks. For stability, we employ backward transfer (*BWT*), which computes the average drop of performance at the previous step after learning the current dataset. In this case, higher appearance

---

[3]We downsample the original VGGSound dataset into a small version, denoted as VGGSound-S.

Table 2: Average performance (mean±std.) on classification and retrieval tasks. Here "R@$k$" is the abbreviation of "Recall@$k$". The results are calculated over ten independent trials. The best result in each case is marked as bold.

| Method | | Classification | | Retrieval | | | |
| | | Acc | $\text{BWT}_A$ | R@1 | R@5 | R@10 | $\text{BWT}_{R10}$ |
|---|---|---|---|---|---|---|---|
| ImageBind | Vanilla | 47.32 ±0.27 | -5.72 ±0.44 | 9.93 ±0.09 | 27.86 ±0.18 | 38.56 ±0.14 | -3.34 ±0.09 |
| | GEM | 40.98 ±1.65 | -9.75 ±1.64 | 9.53 ±0.14 | 27.00 ±0.28 | 37.38 ±0.31 | -2.91 ±0.26 |
| | DER | 47.34 ±0.41 | -5.70 ±0.54 | 9.91 ±0.08 | 27.88 ±0.18 | 38.55 ±0.20 | -3.32 ±0.08 |
| | DER++ | 47.40 ±0.15 | -4.71 ±0.48 | 9.94 ±0.12 | 27.95 ±0.18 | 38.72 ±0.13 | -3.09 ±0.11 |
| | EWC | 48.34 ±1.53 | -4.82 ±1.18 | 10.07 ±0.24 | 28.35 ±0.55 | 39.22 ±0.69 | -2.55 ±0.76 |
| | Co$^2$L | 50.13 ±0.52 | -3.74 ±0.51 | 9.79 ±0.04 | 27.99 ±0.21 | 38.66 ±0.23 | -1.77 ±0.19 |
| | C-FLAT | 50.96 ±0.34 | -4.17 ±0.48 | 9.92 ±0.09 | 28.90 ±0.14 | 39.97 ±0.19 | -1.41 ±0.19 |
| | CILA | 50.73 ±0.48 | -3.20 ±0.74 | 9.68 ±0.11 | 27.76 ±0.20 | 38.25 ±0.20 | -1.28 ±0.21 |
| | DNS (ours) | **52.52** ±0.23 | **-0.02** ±0.30 | **10.38** ±0.09 | **29.53** ±0.17 | **40.89** ±0.10 | **-1.07** ±0.12 |
| LanguageBind | Vanilla | 51.86 ±1.13 | -15.71 ±1.22 | 7.51 ±0.10 | 25.36 ±0.32 | 36.02 ±0.49 | -10.19 ±0.53 |
| | GEM | 60.15 ±0.63 | -4.98 ±0.52 | 8.32 ±0.13 | 28.92 ±0.14 | 41.24 ±0.22 | -4.12 ±0.18 |
| | DER | 52.55 ±1.08 | -14.96 ±0.88 | 7.55 ±0.16 | 25.47 ±0.36 | 36.08 ±0.46 | -10.11 ±0.46 |
| | DER++ | 55.86 ±1.64 | -10.45 ±1.58 | 7.33 ±0.20 | 25.66 ±0.53 | 36.50 ±0.61 | -9.58 ±0.62 |
| | EWC | 54.03 ±1.94 | -13.45 ±2.34 | 8.00 ±0.26 | 27.11 ±0.92 | 38.14 ±1.26 | -7.71 ±1.44 |
| | Co$^2$L | 58.08 ±0.64 | -5.79 ±1.22 | 8.33 ±0.12 | 28.33 ±0.24 | 40.15 ±0.31 | -3.92 ±0.37 |
| | C-FLAT | 57.84 ±0.75 | -7.57 ±0.92 | 8.23 ±0.10 | 28.02 ±0.17 | 39.72 ±0.25 | -5.00 ±0.32 |
| | CILA | 59.30 ±0.62 | -2.63 ±0.91 | 8.32 ±0.14 | 28.31 ±0.26 | 40.48 ±0.27 | **-1.09** ±0.40 |
| | DNS (ours) | **64.07** ±0.37 | **-0.09** ±0.59 | **8.71** ±0.06 | **29.81** ±0.23 | **42.44** ±0.20 | -3.00 ±0.21 |
| UniBind | Vanilla | 47.48 ±0.56 | -5.46 ±0.61 | 10.09 ±0.10 | 28.16 ±0.26 | 38.93 ±0.24 | -3.52 ±0.20 |
| | GEM | 41.37 ±1.52 | -9.29 ±1.20 | 9.61 ±0.14 | 27.32 ±0.31 | 37.94 ±0.34 | -2.97 ±0.39 |
| | DER | 47.64 ±0.57 | -5.35 ±0.65 | 10.10 ±0.09 | 28.16 ±0.25 | 38.95 ±0.26 | -3.51 ±0.20 |
| | DER++ | 47.75 ±0.23 | -4.69 ±0.71 | 10.06 ±0.08 | 28.30 ±0.21 | 39.10 ±0.15 | -3.39 ±0.12 |
| | EWC | 48.21 ±1.48 | -5.06 ±1.19 | 10.24 ±0.25 | 28.60 ±0.54 | 39.47 ±0.66 | -2.89 ±0.67 |
| | Co$^2$L | 50.45 ±0.59 | -3.47 ±0.92 | 9.87 ±0.08 | 28.30 ±0.16 | 39.21 ±0.20 | -1.87 ±0.21 |
| | C-FLAT | 51.25 ±0.42 | -4.36 ±0.49 | 10.00 ±0.06 | 29.19 ±0.17 | 40.51 ±0.12 | -1.48 ±0.18 |
| | CILA | 51.03 ±0.41 | -2.93 ±0.75 | 9.76 ±0.08 | 28.07 ±0.14 | 38.80 ±0.17 | -1.40 ±0.20 |
| | DNS (ours) | **52.86** ±0.29 | **0.31** ±0.46 | **10.52** ±0.06 | **29.97** ±0.10 | **41.44** ±0.11 | **-1.19** ±0.12 |

scores of these metrics indicate higher consistency and determinism. More details of the metrics can be checked in Appendix A.5.

## 4.2 Results

**Overall evaluation.** We compare our method (DNS) with selected baselines on both classification and retrieval tasks with results being reported in Table 2. This result demonstrates that DNS outperforms all baselines on two types of performance metrics (*i.e.*, Acc and Recall). Particularly in the classification task, DNS (ours) consistently achieves the highest accuracy and exhibits minimal or even positive $\text{BWT}_A$, whereas other methods generally suffer from higher negative transfer. In the retrieval task, DNS outperforms other methods in all Recall@$k$ (R@1, R@5, and R@10) metrics, and it also demonstrates good backward transfer ability. These results highlight the effectiveness of DNS in mitigating catastrophic forgetting (stability) and achieving superior performance (plasticity) across both classification and retrieval tasks.

**Stability analyses.** We calculate the deviations between alignment scores to provide practical support for the theoretical guarantee of stability in Theorem 5. At each step, we average the scores of modality pairs, which will be compared at subsequent steps, with the results shown in Figure 4 (a). We can observe that the deviations are small and even reach nearly zero at some steps, further sustaining the stability maintaining of DNS.

**Plasticity analyses.** We provide empirical support corresponding to the theoretical insights on plasticity in Theorem 6. Here we compute the high-order term and verify whether it satisfies $\frac{o(\eta)}{\eta} \leq 0$. We report the results in Figure 4 (b) and have the following observations. At each step $t > 1$,

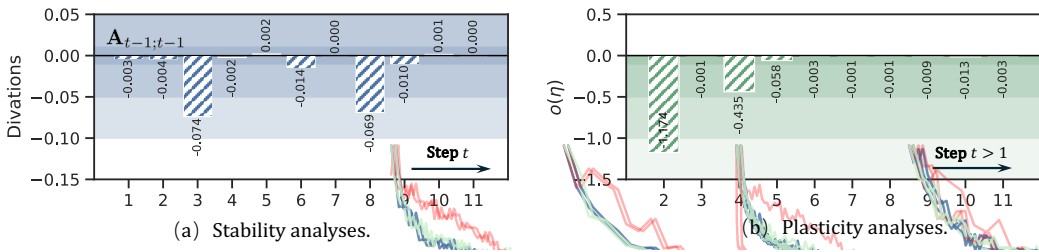

Figure 4: The illustrations on stability and plasticity: the left (a) showcases deviations in the alignment score from $\mathbf{A}_{t-1;t-1}$; the right (b) represents the average of high-order terms for step $t > 1$.

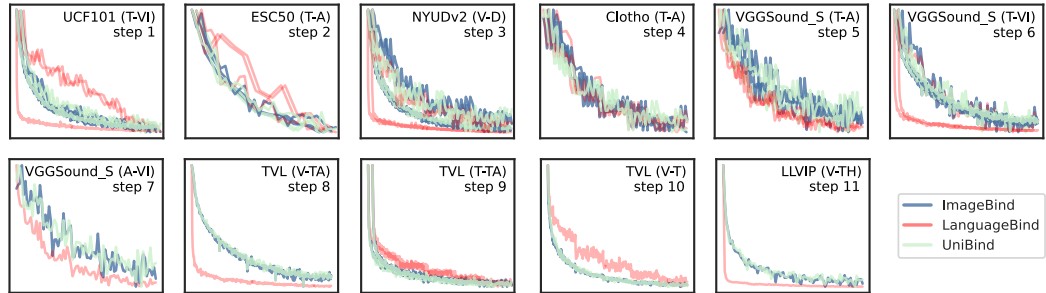

Figure 5: The curves plot batch-wise training losses *w.r.t.* different modality-binding methods.

the averaged $o(\eta)$ is negative and close to 0, which confirms our claim in Theorem 6 in a practical scenario. Furthermore, the training loss converging supports the plasticity as well, in Section 4.3.

## 4.3 More Analyses and Justifications

**Training loss curves.** We evaluate the stability and plasticity through training losses over batches. The batch-wise training loss illustrates the learning status at the own step (plasticity). The results are showcased in Figure 5. From the observation, our method can achieve a stable training loss even after the model is optimized by new datasets, and the batch-wise loss also successfully converges. These findings demonstrate that DNS reaches a balance between stability and plasticity.

**Imbalanced modality data.** We construct a specialized dataset with modality pairs of text-video ($t_0$) and text-audio ($t_1, t_2, t_3$), to assess performance in imbalanced scenarios. We test text-video at step 1 (with less overlap) and text-audio at step 2 (with the most overlap), where greater overlap indicates worse forgetting and larger performance

|  | **Best** | **Vanilla** | **CILA** | **DNS** |
|---|---|---|---|---|
| $t_0$ (T-VI) | 87.72 | 85.34 | 85.54 | 87.92 |
| $t_1$ (T-A) | 49.25 | 41.00 | 43.75 | 47.50 |

Table 3: Performance comparison under imbalanced modality data scenarios.

drops. As shown in the Table 3, DNS consistently outperforms Vanilla and CILA, demonstrating superior robustness.

**Efficiency analyses.** DNS maintains efficiency for CMCL, which differs from replay-based methods that require additional buffer memory and further introduce more computational costs. Here we evaluate the efficiency of our methods by recording the time in training, as shown in Figure 6. The replay-based method requires more training time compared to both the vanilla method and DNS. DNS introduces a slight increase in training time due to the gradient projection and SVD approximation (less than 1s), while still preserving overall training efficiency.

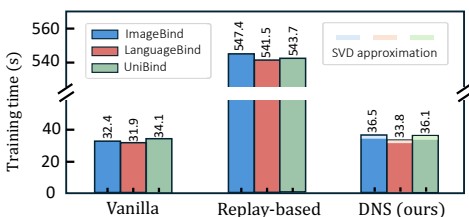

Figure 6: The time recording (s) for vanilla, replay (DER), and DNS training for one step.

**Misaligned & noisy pairs.** We create misaligned modality pairs by exchanging features at ratios of $\{0.01, 0.05, 0.1, 0.2, 0.3, 0.4, 0.5\}$. The results are shown in Table 4. As misalignment increases, performance decreases across methods. However, DNS consistently outperforms CILA, demonstrating stronger robustness. We also add Gaussian noise to the training dataset at scales of

| ratio | Acc | | BWT$_A$ | | Recall | | BWT$_R$ | | Acc | | BWT$_A$ | | Recall | | BWT$_R$ | |
| | *Misalignment* | | | | | | | | *Noise* | | | | | | | |
| | CILA | DNS | CILA | DNS | CILA | DNS | CILA | DNS | CILA | DNS | CILA | DNS | CILA | DNS | CILA | DNS |
|---|---|---|---|---|---|---|---|---|---|---|---|---|---|---|---|---|
| 0.01 | 49.39 | 52.04 | -3.52 | 0.11 | 38.03 | 40.32 | -1.72 | -1.35 | 50.37 | 52.18 | -3.12 | -0.40 | 38.46 | 40.65 | -1.25 | -1.11 |
| 0.05 | 46.45 | 46.98 | -1.81 | 2.93 | 37.46 | 39.41 | -1.32 | -1.22 | 47.42 | 50.74 | -5.31 | -1.79 | 34.59 | 39.79 | -4.41 | -2.66 |
| 0.1 | 41.46 | 43.54 | 0.36 | 2.00 | 36.21 | 38.50 | -0.90 | -0.90 | 43.89 | 50.44 | -7.94 | -0.69 | 34.19 | 40.21 | -4.07 | -0.66 |
| 0.2 | 33.33 | 35.18 | 1.26 | 1.39 | 33.91 | 36.63 | -1.49 | -1.85 | 39.60 | 43.82 | -9.54 | -6.32 | 32.58 | 37.29 | -5.09 | -3.56 |
| 0.3 | 28.73 | 29.27 | -0.80 | 1.46 | 32.12 | 34.09 | -1.94 | -2.32 | 34.45 | 39.48 | -13.80 | -11.15 | 30.62 | 35.77 | -7.41 | -4.30 |
| 0.4 | 19.91 | 24.97 | -5.59 | 0.17 | 29.43 | 32.84 | -2.60 | -2.64 | 31.92 | 39.22 | -14.36 | -11.63 | 29.30 | 34.89 | -8.53 | -5.26 |
| 0.5 | 17.32 | 19.78 | -6.48 | -4.30 | 27.58 | 30.55 | -3.86 | -4.03 | 28.52 | 37.31 | -18.62 | -11.29 | 28.70 | 33.35 | -7.94 | -6.72 |

Table 4: Performance comparison under misaligned modality pairs (left) and noisy pairs (right).

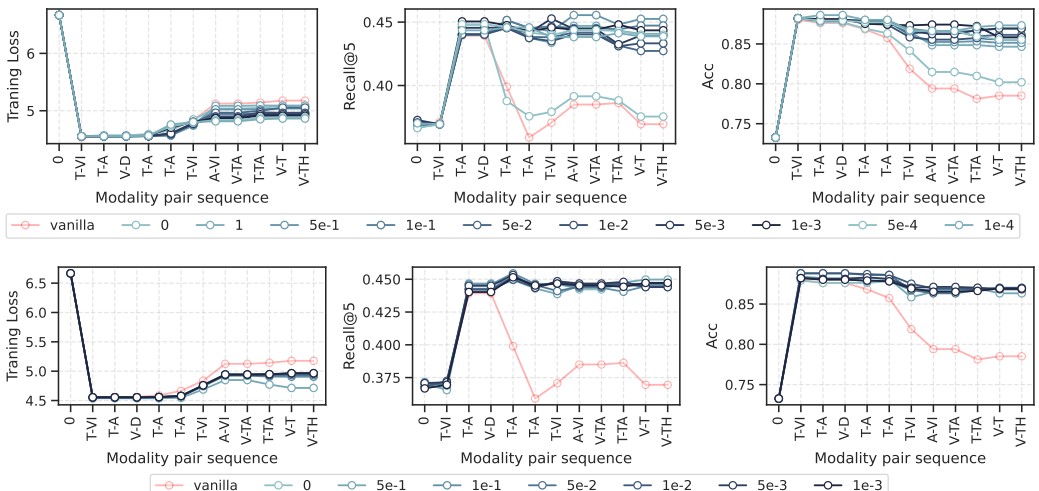

Figure 7: Detailed training loss (*left*), recall@5 (*middle*), and accuracy (*right*) results on the data at the first step. The top figure indicates the results for different $\lambda_{\min}$, while the bottom is for different weight decays, with *ImageBind* as the backbone. The red line indicates the performance of the vanilla method in CMCL.

$\{0.01, 0.05, 0.1, 0.2, 0.3, 0.4, 0.5\}$. Performance decreases with increasing noise, but DNS outperforms CILA across all metrics, showcasing greater robustness.

**Hyperparameter analysis.** We provide detialed results on the step-wise training loss and performance changes on the data at the first step shown in Figure 7 with ImageBind as the backbone. Compared to the vanilla method, DNS maintains a stable performance with continual data training. We also provide empirical results on other models in Figures 8, and 9, under different hyperparameters settings (*cf.*, Appendix F). More than that, we report the detailed variations at each step in Figures 10, 11, and 12 (*cf.*, Appendix F).

## 5  Conclusion

In this paper, we introduce and formally define continual multimodal contrastive learning (CMCL), a research question on how to incrementally integrate multimodal data through contrastive learning. We specify the definitions of stability and plasticity in CMCL. Theoretically, we derive a novel method, termed DNS, which employs a dual-sided gradient-projection strategy to balance the incorporation of new modality pairs while preserving existing multimodal knowledge effectively. The parameter gradients are projected from one side of its own modality and another side of the interacted modalities onto subspaces. Any newly-projected gradient within these subspaces makes no effect on the prior effective parameter spaces. Furthermore, two upper bounds are provided with theoretical insights on both stability and plasticity, and support the effectiveness of our gradient-projection method. Empirical results across seven datasets demonstrate our method's superiority over current advanced continual learning baselines, which confirm its robustness and efficiency.

## Acknowledgements

This research/project is supported by the National Research Foundation, Singapore under its National Large Language Models Funding Initiative (AISG Award No: AISG-NMLP2024-002). Any opinions, findings and conclusions or recommendations expressed in this material are those of the author(s) and do not reflect the views of National Research Foundation, Singapore.

Xiaobo Xia is also supported by MoE Key Laboratory of Brain-inspired Intelligent Perception and Cognition, University of Science and Technology of China (Grant No. 2421002).

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

# Appendix

# A  Implementation Details

## A.1  Training and Inference Procedures

Here we provide the training algorithm flow in Algorithm 1 and the corresponding pseudo-code. The inference algorithm flow is demonstrated in Algorithm 2. Algorithm 1 describes the Continual Multimodal Contrastive Learning training process with our method DNS, which enables a model to progressively learn from new multimodal data while maintaining stability in previously learned modality alignments.

---

**Algorithm 1** DNS for Continual Multimodal Contrastive Learning (Training)

---

**Require:  Inputs:**

    Pre-trained encoders $\{g_m^*\}_{m \in \mathcal{M}}$ and follow-up learner $\{\mathbf{W}^m\}_{m \in \mathcal{M}}$ for each modality $m$, learning rate $\eta$,

    A sequence of training datasets $\{\mathcal{X}_t^{m,m'}\}_{t=1}^N$, each containing modality pairs.

    Feature covariance matrix $\{\bar{\mathbf{Z}}_{<t;*}^m\}_{m \in \mathcal{M}}$ and $\{\bar{\mathbf{Z}}_{<t;<t}^m\}_{m \in \mathcal{M}}$ (initially $t = 1$).

**Ensure:**  Updated model parameters $\{\mathbf{W}_t^m\}$ (*i.e.*, $\{\theta_m^t\}$) that preserve alignment stability on previously seen data.

1: **Initialize** features covariance storages $\{\bar{\mathbf{Z}}_{<t;*}^m\}_{m \in \mathcal{M}}$ and $\{\bar{\mathbf{Z}}_{<t;<t}^m\}_{m \in \mathcal{M}}$ to zero.

2: **Initialize** local parameters from $\{\mathbf{W}^m\}_{m \in \mathcal{M}}$.

3: **for** $t = 1$ to $N$ **do**

4:    **Prior to the new step** $\{m, m'\}$**:**

5:       ① Retrieve feature variance $\{\bar{\mathbf{Z}}_{<t;*}^m\}_{m \in \{m,m'\}}$ and $\{\bar{\mathbf{Z}}_{<t;<t}^m\}_{m \in \{m,m'\}}$.

6:       ② Compute the projectors:

7:

$$\mathbf{P}_{<t} \leftarrow \bar{\mathbf{Z}}_{<t;*}^m, \quad \tilde{\mathbf{P}}_{<t} \leftarrow \bar{\mathbf{Z}}_{<t;<t}^m, \quad \mathbf{P}'_{<t} \leftarrow \bar{\mathbf{Z}}_{<t;*}^{m'}, \quad \tilde{\mathbf{P}}'_{<t} \leftarrow \bar{\mathbf{Z}}_{<t;<t}^{m'}$$

8:    **Train on the new dataset** $\mathcal{X}_t^{m,m'}$:

9:      **Repeat** until convergence:

10:        ① Sample a mini-batch $\{(\mathbf{x}^m, \mathbf{x}^{m'})\}$ from $\mathcal{X}_t^{m,m'}$.

11:        ② Compute representations and the multimodal contrastive loss:

$$\mathcal{L}_{m,m'} = \mathcal{L}\big(g_m(\mathbf{x}^m; \mathbf{W}_{t-1}^m), \; g_{m'}(\mathbf{x}^{m'}; \mathbf{W}_{t-1}^{m'})\big).$$

12:        ③ Compute the raw gradients:

$$\nabla \mathbf{W}_t^m \leftarrow \frac{\partial \mathcal{L}_{m,m'}}{\partial \mathbf{W}_{t-1}^m}, \quad \nabla \mathbf{W}_t^{m'} \leftarrow \frac{\partial \mathcal{L}_{m,m'}}{\partial \mathbf{W}_{t-1}^{m'}}.$$

13:        ④ **Project the gradients** so that old alignments are preserved (Theorem 4):

$$\Delta \mathbf{W}_t^m \;=\; \nabla \mathbf{W}_t^m \;-\; \tilde{\mathbf{P}}_{<t}\, \nabla \mathbf{W}_t^m\, \mathbf{P}'_{<t},$$

$$\Delta \mathbf{W}_t^{m'} \;=\; \nabla \mathbf{W}_t^m \;-\; \tilde{\mathbf{P}}'_{<t}\, \nabla \mathbf{W}_t^m\, \mathbf{P}_{<t},$$

14:        ⑤ **Update the parameters**:

$$\mathbf{W}_t^m \;\leftarrow\; \mathbf{W}_{t-1}^m \;-\; \eta\, \Delta \mathbf{W}_t^m, \quad \mathbf{W}_t^{m'} \;\leftarrow\; \mathbf{W}_{t-1}^{m'} \;-\; \eta\, \Delta \mathbf{W}_t^{m'}.$$

15:      End mini-batch.

16:    **Update the feature covariances (step-wise or batch-wise):**

17:      $\bar{\mathbf{Z}}_{<t+1;*}^m = \frac{\bar{n}_{t-1}}{\bar{n}_t} \bar{\mathbf{Z}}_{<t;*}^m + \frac{1}{\bar{n}_t} (\mathbf{Z}_{t;*}^m)^\top \mathbf{Z}_{t;*}^m, \quad \bar{\mathbf{Z}}_{<t+1;<t+1}^m = \frac{\bar{n}_{t-1}}{\bar{n}_t} \bar{\mathbf{Z}}_{<t;<t}^{m'} + \frac{1}{\bar{n}_t} (\mathbf{Z}_{t;t}^{m'})^\top \mathbf{Z}_{t;t}^{m'},$

18:      $\bar{\mathbf{Z}}_{<t+1;*}^{m'} = \frac{\bar{n}_{t-1}}{\bar{n}_t} \bar{\mathbf{Z}}_{<t;*}^{m'} + \frac{1}{\bar{n}_t} (\mathbf{Z}_{t;*}^{m'})^\top \mathbf{Z}_{t;*}^{m'}, \quad \bar{\mathbf{Z}}_{<t+1;<t+1}^{m'} = \frac{\bar{n}_{t-1}}{\bar{n}_t} \bar{\mathbf{Z}}_{<t;<t}^m + \frac{1}{\bar{n}_t} (\mathbf{Z}_{t;t}^m)^\top \mathbf{Z}_{t;t}^m.$

19:

20: **end for**

21: **Return:** $\{\mathbf{W}_N^m\}$ preserving stability for all previously learned pairs $(m, m')$.

---

DNS provides a convenient way to modify gradients before the optimizer step. By applying structured transformations using precomputed projection matrices, it ensures better optimization while keeping the implementation modular and flexible. The pseudo-code for gradient updates is shown below.

---

**Pseudo-code for gradient updates**

```
# ...
def update_gradients(self):
    # retrieve the gradients ...
    gradient1 -=  self.P_1_ @ gradient1 @ self.P_1
    gradient2 -=  self.P_2_ @ gradient2 @ self.P_2
    # overwrite the gradients ...

# ...
# invoke the function before backward propagation
model.update_gradients()
optimizer.step()
```

---

Algorithm 2 describes the inference process for Continual Multimodal Contrastive Learning. The goal is to perform inference on a given test sample from a specific modality and generate aligned representations.

---

**Algorithm 2** DNS for Continual Multimodal Contrastive Learning (Inference)

---

**Require: Inputs:**
   Pre-trained encoders $\{g_m^*\}_{m \in \mathcal{M}}$ and trained learner $\{\mathbf{W}^m\}_{m \in \mathcal{M}}$ for each modality $m$,
   A test sample (or a query) in modality $m$, denoted $\mathbf{x}^m$.
**Ensure:** Inference outputs (e.g., aligned representations, retrieval scores).
 1: **Obtain representation**:

$$\mathbf{z}^m = \mathbf{W}_N^m g_m(\mathbf{x}^m), \quad \mathbf{z}^{m'} = \mathbf{W}_N^{m'} g_{m'}(\mathbf{x}^{m'}).$$

 2: **Perform downstream task**:
 3:    For example, retrieve the best match in another modality $m'$ via

$$\max_{\mathbf{z}^{m'}} (\mathbf{z}^m)^\top \mathbf{z}^{m'} \quad \text{(or apply other alignment criteria).}$$

 4: **Output** the alignment scores, predicted labels, or retrieved results, *etc*.

---

## A.2   Datasets

Below, we introduce the used datasets in more detail.

- **UCF101** [29][4] is a widely used dataset for action recognition across 101 action categories, sourced from YouTube. It extends the UCF50 dataset, introducing greater diversity and real-world complexity with significant variations in camera motion, object appearance, viewpoint, background clutter, and illumination. The 101 action categories span 5 broad types: 1) Human-Object Interaction (*e.g.,* cutting in kitchen); 2) Body-Motion Only (*e.g.,* jump rope and Tai Chi); 3) Human-Human Interaction (*e.g.,* boxing and head massage); 4) Playing Musical Instruments (*e.g.,* playing violin and playing tabla); 5) Sports (*e.g.,* basketball dunk and pole vault). In our case, we select 70 examples for each category for training, and 10 examples for each category for testing to balance the distribution.

---

[4]https://www.crcv.ucf.edu/data/UCF101.php

- **ESC50** [30][5] is a collection of 2,000 five-second environmental audio recordings, designed for benchmarking methods in environmental sound classification. It consists of 50 semantically distinct classes, each containing 40 examples, grouped into 5 major categories: 1) Animals (*e.g.,* dog, cat, and frog); 2) Natural Soundscapes & Water Sounds (*e.g.,* rain, sea waves, and thunderstorm); 3) Human Non-Speech Sounds (*e.g.,* laughing, sneezing, and footsteps); 4) Interior/Domestic Sounds (*e.g.,* door knock, washing machine, and clock alarm); 5) Exterior/Urban Noises (*e.g.,* car horn, airplane, and fireworks). All clips have been manually extracted from public field recordings available on Freesound.org. To evaluate the performance, we randomly split the data with a ratio of 4:1 for training (1,600) and testing (400).

- **NYUDv2** [31, 42][6] is a dataset originally for depth estimation and semantic segmentation, which features video sequences of indoor scenes captured with both RGB and depth cameras from Microsoft Kinect. We use the preprocessed version of the original NYU Depth V2 dataset linked above. It is also used in the TensorFlow dataset[7]. In this version, 47,584 examples are used for training, and 654 examples are for testing.

- **VGGSound-S** [32][8] is extracted from content-sharing platform, YouTube, consisting of short clips of audio sounds. It consists of 10-second video segments collected "in the wild", which ensures that the sound source is visually evident in each clip. The dataset spans a wide range of acoustic environments and diverse noise conditions, making it highly representative of real-world applications. To facilitate the training and evaluation, we downsample a small version of VGGSound, namely VGGSound-S, which consists of 10,000 examples for training and 2,000 examples for testing.

- **Clotho** [33][9] is an audio captioning dataset, with all audio examples being from the Freesound platform. The duration of the audio is in the range of 15 to 30 seconds. Each audio has captions up to 20 words in length, collected by AMT or a specific protocol for crowd-sourcing audio annotations. We only keep one caption per audio in our case and use its development set (3,840) for training and evaluation set (1,045) for testing.

- **TVL** [34][10] is a large-scale image-touch dataset designed for multimodal learning, combining two datasets: 1) SSVTP [43], a robot-collected dataset with 4,587 image-touch pairs, manually labeled with image annotations; and 2) HCT, a large-scale dataset with 39,154 synchronously captured in-the-wild image-touch pairs, labeled using GPT-4V [44] based on visual inputs. We use its provided split for training and testing.

- **LLVIP** [35][11] is a visible-infrared paired dataset designed for low-light vision applications, which contains 30,976 images (15,488 pairs) captured in extremely dark scenes. All images are strictly aligned in time and space. We also use its default split for training and testing.

## A.3 Modality-Binding Methods

Table 5: Details of different modality-binding methods, which include the supported modalities (V↔Vision, T↔Text, VI↔VIdeo, A↔Audio, D↔Depth, TH↔THermal, IMU↔Inertial Measurement Unit, EV↔EVent, and P↔Point cloud), feature dimension, and the number of datasets for pre-training.

| | Modalities | Feature dimension | #Datasets |
|---|---|---|---|
| **ImageBind** | V, T, VI, A, D, TH, IMU | 1,024 | 4 |
| **LanguageBind** | V, T, VI, A, D, TH | 768 | 6 |
| **UniBind** | V, T, VI, A, TH, EV, P | 1,024 | 13 |

---

[5]https://github.com/karolpiczak/ESC-50

[6]https://cs.nyu.edu/~fergus/datasets/nyu_depth_v2.html

[7]https://www.tensorflow.org/datasets/catalog/nyu_depth_v2

[8]https://www.robots.ox.ac.uk/~vgg/data/vggsound/

[9]https://zenodo.org/records/3490684

[10]https://tactile-vlm.github.io/

[11]https://bupt-ai-cz.github.io/LLVIP/

Here we introduce three modality-binding methods used in this work (also check the details in Table 5).

- **ImageBind** [11][12] aligns six modalities, *i.e.*, images, text, audio, depth, thermal, and IMU, into a joint embedding space, with vision-relevant modality pair data. It utilizes 4 datasets, containing video-audio (Audioset [45]), image-depth (SUN RGB-D [46]), image-thermal (LLVIP [35]), and video-IMU (Ego4D [47]), for pre-training.

- **LanguageBind** [15][13] takes language (*i.e.*, text) as the bind across different modalities, including video, infrared, depth, audio and text. Languagebind is pre-trained on the collected dataset VIDAL-10M [15], which is extracted from 6 datasets: YouTube-8M [48], MSR-VTT [49], COCO [50], AVA [51], HMDB-51 [52], and ImageNet [53].

- **UniBind** [13][14] learns a unified and balanced representation space with large language models (LLMs) for data augmentation. All modality embeddings are aligned with the embedding centers yielded from LLM-augmented data. 1,000 descriptions for each category are generated via LLMs (GPT-4 [44] and LLaMA [54]), and multimodal data descriptions are obtained by multimodal LLMs like BLIP2 [55] and LLaMA-Adapter [56].

## A.4 Baselines

Below, we discuss the used baselines in more detail.

- **GEM** [38] utilizes an episodic memory to store observed examples for each task. GEM computes gradients on the current task and compares them to gradients on the stored episodic memory. The inner product between these loss gradient vectors is assumed to be positive ($\langle \nabla_t, \nabla_k \rangle \geq 0, \forall k < t$) so that the proposed parameter update $\nabla_t$ is unlikely to increase the loss at the previous tasks.

- **DER & DER++** [39] maintain the network's logits from the past tasks with being compared to the logits at the current task. In particular, DER++ further introduces the ground-truth labels serving as an additional regularization to stabilize the optimization. In our case, since we do not have explicit labels, we directly use a binary class to determine whether the modality pairs are from the same instance.

- **EWC** [40] utilizes the Fisher information matrix to estimate the importance of each parameter from previous tasks. Besides, EWC penalizes significant deviations from previous optimal values. As a result, it adds a quadratic constraint term to the loss function. The posterior is approximated as a Gaussian distribution and resolved via a Laplace approximation.

- **$Co^2L$** [17] is a replay-based continual contrastive learning method. In more detail, $Co^2L$ modifies the supervised contrastive loss by viewing the examples within current tasks as the anchor, while examples at past tasks as the negatives. The instance-wise relations from the previous model are distilled to preserve the learned knowledge. Therefore, two objectives are combined with a controllable hyperparameter for the optimization of model parameters.

- **C-FLAT** [41] introduces an optimization method to enhance the existing continual learning method by promoting loss landscape flatness. It incorporates zeroth-order and first-order sharpness-aware optimization, balancing both local minima selection and landscape smoothness. C-FLAT is also encapsulated as a "plug-and-play" optimizer. In our case, we implement this method built upon $Co^2L$.

- **CILA** [18] decomposes the continual learning objective into two components: a supervised contrastive loss and a distillation loss, following $Co^2L$. In particular, CILA introduces an adaptive distillation induced by theoretical analyses. The distillation coefficients are dynamically updated based on the ratio of past distillation and contrastive losses.

---

[12]https://github.com/facebookresearch/ImageBind
[13]https://github.com/PKU-YuanGroup/LanguageBind
[14]https://github.com/QC-LY/UniBind

We implement these methods following public repositories like mammoth[15], Co$^2$L[16], and GradientEpisodicMemory[17].

## A.5 Evaluation Settings

**Evaluation metrics.** We use Recall@$k$, Acc, and BWT for retrieval and classification tasks. BWT indicates the anti-forgetting capability. Recall@$k$ measures the ability of a model to retrieve the correct item among the top-$k$ candidates.

Formally, for a dataset with $n$ queries, Recall@$k$ is defined as:

$$\text{Recall@}k := \frac{1}{n}\sum_{i=1}^{n}\mathbb{I}(r(i) \leq k), \tag{11}$$

where $\mathbb{I}(\cdot)$ is the indicator function that returns 1 if the condition is true and 0 otherwise; $r(\cdot)$ denotes the position of the ground truth item for the $i$-th query in the ranked retrieval list. For classification tasks, average accuracy (Acc) is employed to measure the proportion of correctly predicted labels. Specifically, given the ground-truth label $y_i$ and the predicted label $\hat{y}_i$ for each instance in datasets with size of $n$, accuracy is given by:

$$\text{Acc} := \frac{1}{n}\sum_{i=1}^{n}\mathbb{I}(y_i = \hat{y}_i). \tag{12}$$

Here $\hat{y}_i$ is calculated by finding the closest categories vectors with the query vector, *i.e.*, $(\mathbf{z}_i^m)^\top \mathbf{z}_c^\mathrm{T}$, where $\mathbf{z}_c^\mathrm{T}$ represents the textual category vector. BWT measures the effect of learning from new datasets on the performance of previously learned datasets. Let $T$ be the total number of training steps. Different steps involve different datasets. That $R_{t;t}$ is the performance at step $t$ immediately after the model is learned on the $t$-th dataset; $R_{t;T}$ is the performance on $i$-th dataset after leaning at the final step $T$. BWT can be formulated as:

$$\text{BWT} := \frac{1}{T-1}\sum_{t=1}^{T-1}(R_{t;T} - R_{t;t}). \tag{13}$$

Note that for different tasks, we compute corresponding backward transfer metrics, *i.e.*, $\text{BWT}_\mathrm{R}$ for the retrieval task and $\text{BWT}_\mathrm{A}$ for the classification task.

**Hyperparameter settings.** We elaborate on the hyperparameter settings to facilitate reproducibility. We employ linear layers for each modality built upon well-trained modality-binding models (see Appendix A.3). This architecture is also similar to UniBind [13]. In practice, we set the same dimension between the input and output features, $\mathbf{W}_1^m \in \mathbb{R}^{d \times d}$, and initialized them as unit matrices. On the one hand, it can directly maintain the original performance from the prior encoders at the initial step. On the other hand, the full-rank property indicates $\text{Col}(\mathbf{Z}_1^m) \subseteq \text{Col}(\mathbf{W}_1^m)$, which further weakens our suppose.

During the training, we employ the Adam optimizer. All models are trained for 5 epochs at every step. The order is consistent for 10 independent trials. In general, we set the learning rate to 0.0001 and the batch size to 64. The weight decay in $\{0, 0.5, 0.1, 0.05, 0.01, 0.005, 0.001\}$. As for the SVD approximation, we set the minimal eigenvalue in the range of $\{0, 0.1, 0.01, 0.001\}$. We provide the hyperparameter analysis with empirical results in Appendix F.

---

[15] https://github.com/aimagelab/mammoth
[16] https://github.com/chaht01/Co2L
[17] https://github.com/facebookresearch/GradientEpisodicMemory

# B  Proofs of Theoretical Results

## B.1  Proof of Theorem 3

> **Recall:**
>
> **Threorem 3.** *Let the global parameter update follow* $(\mathbf{W}_t^{m_1})^\top \mathbf{W}_t^{m_2} = (\mathbf{W}_{t-1}^{m_1})^\top \mathbf{W}_{t-1}^{m_2} - \eta \bar{\mathbf{W}}$, *and the original parameter update be denoted as:*
>
> $$\tilde{\mathbf{W}} = \left( (\mathbf{W}_{t-1}^{m_1})^\top \nabla \mathbf{W}_t^{m_2} + (\nabla \mathbf{W}_t^{m_1})^\top \mathbf{W}_{t-1}^{m_2} - \eta (\nabla \mathbf{W}_t^{m_1})^\top \nabla \mathbf{W}_t^{m_2} \right). \qquad (14)$$
>
> *Then, the stability holds if*
>
> $$\bar{\mathbf{W}} := \tilde{\mathbf{W}} - \mathbf{P}' \tilde{\mathbf{W}} \mathbf{P}, \qquad (15)$$
>
> *where* $\mathbf{P}'$ *and* $\mathbf{P}$ *are the space projectors for* $\mathbf{Z}_{t-1;*}^{m_1}$ *and* $\mathbf{Z}_{t-1;*}^{m_2}$, *respectively.*

*Proof.* Expand $(\mathbf{Z}_{t-1;*}^{m_1})^\top \mathbf{Z}_{t-1;*}^{m_2}$ with the weight updating:

$$(\mathbf{Z}_{t-1;t}^{m_1})^\top \mathbf{Z}_{t-1;t}^{m_2} \qquad (16)$$

$$= (\mathbf{Z}_{t-1;*}^{m_1})^\top \left( \mathbf{W}_{t-1}^{m_1} - \eta \nabla \mathbf{W}_t^{m_1} \right)^\top \left( \mathbf{W}_{t-1}^{m_2} - \eta \nabla \mathbf{W}_t^{m_2} \right) \mathbf{Z}_{t-1;*}^{m_2} \qquad (17)$$

$$= \underbrace{(\mathbf{Z}_{t-1;*}^{m_1})^\top (\mathbf{W}_{t-1}^{m_1})^\top \mathbf{W}_{t-1}^{m_2} \mathbf{Z}_{t-1;*}^{m_2}}_{\mathbf{A}_{t-1;t-1}^{m_1,m_2}} - \eta (\mathbf{Z}_{t-1;*}^{m_1})^\top (\mathbf{W}_{t-1}^{m_1})^\top \nabla \mathbf{W}_t^{m_2} \mathbf{Z}_{t-1;*}^{m_2} \qquad (18)$$

$$- \eta (\mathbf{Z}_{t-1;*}^{m_1})^\top (\nabla \mathbf{W}_t^{m_1})^\top \mathbf{W}_{t-1}^{m_2} \mathbf{Z}_{t-1;*}^{m_2} + \eta^2 (\mathbf{Z}_{t-1;*}^{m_1})^\top (\nabla \mathbf{W}_t^{m_1})^\top \nabla \mathbf{W}_t^{m_2} \mathbf{Z}_{t-1;*}^{m_2} \qquad (19)$$

$$= \mathbf{A}_{t-1;t-1}^{m_1,m_2} - \eta (\mathbf{Z}_{t-1;*}^{m_1})^\top \left( (\mathbf{W}_{t-1}^{m_1})^\top \nabla \mathbf{W}_t^{m_2} + (\nabla \mathbf{W}_t^{m_1})^\top \mathbf{W}_{t-1}^{m_2} - \eta (\nabla \mathbf{W}_t^{m_1})^\top \nabla \mathbf{W}_t^{m_2} \right) \mathbf{Z}_{t-1;*}^{m_2}. \qquad (20)$$

For the stability to hold, the perturbations introduced by the weight updates must not interfere with the alignment. Thus, the following terms must vanish:

$$( \underbrace{\mathbf{Z}_{t-1;*}^{m_1}}_{\mathbf{Z}' \in \mathbb{R}^{d \times N}} )^\top \underbrace{\left( (\mathbf{W}_{t-1}^{m_1})^\top \nabla \mathbf{W}_t^{m_2} + (\nabla \mathbf{W}_t^{m_1})^\top \mathbf{W}_{t-1}^{m_2} - \eta (\nabla \mathbf{W}_t^{m_1})^\top \nabla \mathbf{W}_t^{m_2} \right)}_{\tilde{\mathbf{W}} \in \mathbb{R}^{d \times d}} \underbrace{\mathbf{Z}_{t-1;*}^{m_2}}_{\mathbf{Z} \in \mathbb{R}^{d \times N}} = 0. \qquad (21)$$

> **Question.** How to find the solution that satisfies $\mathbf{Z}'^\top \bar{\mathbf{W}} \mathbf{Z} = 0$ with minimal deviation from the original weight $\tilde{\mathbf{W}}$, *i.e.,* solving:
>
> $$\min_{\bar{\mathbf{W}}} \|\tilde{\mathbf{W}} - \bar{\mathbf{W}}\|, \quad s.t., \quad \mathbf{Z}'^\top \bar{\mathbf{W}} \mathbf{Z} = 0. \qquad (22)$$

① **Find the general solution for** $\mathbf{Z}'^\top \bar{\mathbf{W}} \mathbf{Z} = 0$.

According to the Lemma 7, we can solve the equation $\mathbf{Z}'^\top \bar{\mathbf{W}} \mathbf{Z} = 0$ and obtain its general solution:

$$\bar{\mathbf{W}} = \underbrace{\mathbf{0}}_{\text{trivial}} + \underbrace{\mathbf{Y} - \mathbf{Z}'^{\top\dagger} \mathbf{Z}'^\top \mathbf{Y} \mathbf{Z} \mathbf{Z}^\dagger}_{\text{non-trivial}} = \mathbf{Y} - (\mathbf{Z}' \mathbf{Z}'^\dagger)^\top \mathbf{Y} \mathbf{Z} \mathbf{Z}^\dagger = \mathbf{Y} - (\mathbf{P}')^\top \mathbf{Y} \mathbf{P} = \mathbf{Y} - \mathbf{P}' \mathbf{Y} \mathbf{P},$$

$$(23)$$

where $\mathbf{Y}$ is arbitrary, $\mathbf{P}'$ and $\mathbf{P}$ are the col-space projectors of $\mathbf{Z}'$ and $\mathbf{Z}$, respectively.

Any element $\bar{\mathbf{W}}$ in set $\{\mathbf{Y} - \mathbf{P}' \mathbf{Y} \mathbf{P} | \mathbf{Y} \in \mathbb{R}^{d \times d}\}$ satisies $\mathbf{Z}'^\top \bar{\mathbf{W}} \mathbf{Z} = 0$.

② **Find the optimal solution for** $\mathbf{Z}'^\top \bar{\mathbf{W}} \mathbf{Z} = 0$. Now given the set:

$$S = \{\mathbf{Y} - \mathbf{P}' \mathbf{Y} \mathbf{P} \mid \mathbf{Y} \in \mathbb{R}^{d \times d}\}, \qquad (24)$$

where $\mathbf{P} \in \mathbb{R}^{d \times d}$ and $\mathbf{P}' \in \mathbb{R}^{d \times d}$ are projection matrices, (*i.e.*, $\mathbf{P}^2 = \mathbf{P}$ and $\mathbf{P}'^2 = \mathbf{P}'$); and their L2 norms are either 1 or 0 (see Lemma 4). Given the matrix $\tilde{\mathbf{W}} \in \mathbb{R}^{d \times d}$, we hope to find the *closest* element in set $S$, in other words, we need to solve the following equation:

$$\min_{\mathbf{Y}} \|\mathbf{Y} - \mathbf{P}'\mathbf{Y}\mathbf{P} - \tilde{\mathbf{W}}\|_F^2. \tag{25}$$

Here, we verify the *convexity* of the above objective function.

The expression for $f(\mathbf{Y})$ ca be written as:

$$f(\mathbf{Y}) = \|\mathbf{Y} - \mathbf{P}'\mathbf{Y}\mathbf{P} - \tilde{\mathbf{W}}\|_F^2 = \operatorname{tr}\left((\mathbf{Y} - \mathbf{P}'\mathbf{Y}\mathbf{P} - \tilde{\mathbf{W}})^\top (\mathbf{Y} - \mathbf{P}'\mathbf{Y}\mathbf{P} - \tilde{\mathbf{W}})\right). \tag{26}$$

$\|\cdot\|_F$ denote the Frobenius norm. The gradient of $f(Y)$ is given by:

$$\nabla_{\mathbf{Y}} f(\mathbf{Y}) = 2\left(\mathbf{Y} - \mathbf{P}'\mathbf{Y}\mathbf{P} - \tilde{\mathbf{W}} - \mathbf{P}'(\mathbf{Y} - \mathbf{P}'\mathbf{Y}\mathbf{P} - \tilde{\mathbf{W}})\mathbf{P}\right) = 2\left(\mathbf{Y} - \mathbf{P}'\mathbf{Y}\mathbf{P} - \tilde{\mathbf{W}} + \mathbf{P}'\tilde{\mathbf{W}}\mathbf{P}\right). \tag{27}$$

We then vectorize the expression using the properties of the Kronecker product:

$$\operatorname{vec}(\mathbf{P}'\mathbf{Y}\mathbf{P}) = (\mathbf{P}^\top \otimes \mathbf{P}')\operatorname{vec}(\mathbf{Y}). \tag{28}$$

The gradient can be rewritten as:

$$\operatorname{vec}(\nabla_{\mathbf{Y}} f(\mathbf{Y})) = 2\left(\mathbf{I} - \mathbf{P} \otimes \mathbf{P}'\right) \operatorname{vec}(\mathbf{Y} - \tilde{\mathbf{W}}). \tag{29}$$

The second-order derivative (Hessian matrix) is defined as the derivative of the gradient with respect to $\operatorname{vec}(\mathbf{Y})$. Therefore, we have:

$$\mathbf{H} = \frac{\partial \operatorname{vec}(\nabla_{\mathbf{Y}} f(\mathbf{Y}))}{\partial \operatorname{vec}(\mathbf{Y})^\top} = 2\left(\mathbf{I} - \mathbf{P} \otimes \mathbf{P}'\right), \tag{30}$$

where $\mathbf{I}$ is the identity matrix of appropriate dimension matching $\mathbf{Y}$, and $\otimes$ represents the Kronecker product.

Due the the eigenvalues of the projection matrices lie in the set $\{0, 1\}$, the eigenvalues of $\mathbf{P} \otimes \mathbf{P}'$ are either 0 or 1. The corresponding eigenvalue $\lambda_i(\mathbf{H})$ is:

$$\lambda_i(\mathbf{H}) = 2\left(1 - \lambda_j(\mathbf{P} \otimes \mathbf{P}')\right) = \begin{cases} 2(1 - 0) = 2, & \text{if } \lambda_i(\mathbf{P} \otimes \mathbf{P}') = 0, \\ 2(1 - 1) = 0, & \text{if } \lambda_i(\mathbf{P} \otimes \mathbf{P}') = 1. \end{cases} \tag{31}$$

$\lambda_i(\mathbf{P} \otimes \mathbf{P}') \in \{0, 1\}$. Since the eigenvalues of $\mathbf{H}$ are non-negative (0 or 2), $\mathbf{H}$ is positive semidefinite. Therefore, the function $f(\mathbf{Y}) = \|\mathbf{Y} - \mathbf{P}'\mathbf{Y}\mathbf{P} - \tilde{\mathbf{W}}\|_F^2$ is convex.

Set the gradient $\nabla_{\mathbf{Y}} f(\mathbf{Y}) = 0$, which gives the extreme value condition:

$$\mathbf{Y} - \mathbf{P}'\mathbf{Y}\mathbf{P} - \tilde{\mathbf{W}} = \mathbf{P}'(\mathbf{Y} - \mathbf{P}'\mathbf{Y}\mathbf{P} - \tilde{\mathbf{W}})\mathbf{P}. \tag{32}$$

Substitute $\mathbf{Y} = \tilde{\mathbf{W}}$ into the extreme value condition:

$$\mathbf{Y} - \mathbf{P}'\mathbf{Y}\mathbf{P} - \tilde{\mathbf{W}} = \mathbf{W} - \mathbf{P}'\tilde{\mathbf{W}}\mathbf{P} - \tilde{\mathbf{W}} = -\mathbf{P}'\tilde{\mathbf{W}}\mathbf{P} \quad \text{(left-hand side)} \tag{33}$$

$$\mathbf{P}'(\mathbf{Y} - \mathbf{P}'\mathbf{Y}\mathbf{P} - \tilde{\mathbf{W}})\mathbf{P} = \mathbf{P}'(\tilde{\mathbf{W}} - \mathbf{P}'\tilde{\mathbf{W}}\mathbf{P} - \tilde{\mathbf{W}})\mathbf{P} = \mathbf{P}'(-\mathbf{P}'\tilde{\mathbf{W}}\mathbf{P})\mathbf{P} \tag{34}$$

$$= -\mathbf{P}'\mathbf{P}'\tilde{\mathbf{W}}\mathbf{P}\mathbf{P} = -\mathbf{P}'\tilde{\mathbf{W}}\mathbf{P} \quad \text{(right-hand side)} \tag{35}$$

Therefore, the left-hand side equals the right-hand side, confirming that $\mathbf{Y} = \tilde{\mathbf{W}}$ satisfies the extreme value condition, implying $\mathbf{Y} = \tilde{\mathbf{W}}$ is an extreme point, and further the optimal solution ($f(\mathbf{Y})$ is convex). Thus, the closest element in the set $S$ to the matrix $\tilde{\mathbf{W}}$ is: $\bar{\mathbf{W}} := \tilde{\mathbf{W}} - \mathbf{P}'\tilde{\mathbf{W}}\mathbf{P}$. $\quad\square$

## B.2  Proof of Theorem 4

> **Recall:**
>
> **Theorem 4.** *Let the local parameter update follow $\mathbf{W}_t^m = \mathbf{W}_{t-1}^m - \eta \Delta \mathbf{W}_t^m$, the stability holds if*
>
> $$\begin{cases} \Delta \mathbf{W}_t^{m_1} := \nabla \mathbf{W}_t^{m_1} - \tilde{\mathbf{P}} \nabla \mathbf{W}_t^{m_1} \mathbf{P}', \\ \Delta \mathbf{W}_t^{m_2} := \nabla \mathbf{W}_t^{m_2} - \tilde{\mathbf{P}}' \nabla \mathbf{W}_t^{m_2} \mathbf{P}, \end{cases} \quad (36)$$
>
> *where $\tilde{\mathbf{P}}'$ and $\tilde{\mathbf{P}}$ are the space projectors for $\mathbf{Z}_{t-1;t-1}^{m_1}$ and $\mathbf{Z}_{t-1;t-1}^{m_2}$, respectively.*

*Proof.* We restate the problem of how to optimize the model or update the gradients:

> **Question.** How can we update the gradients $\nabla \mathbf{W}_t^{m_1}$ and $\nabla \mathbf{W}_t^{m_2}$ to approximate $\left( (\mathbf{W}_{t-1}^{m_1})^\top \nabla \mathbf{W}_t^{m_2} + (\nabla \mathbf{W}_t^{m_1})^\top \mathbf{W}_{t-1}^{m_2} - \eta (\nabla \mathbf{W}_t^{m_1})^\top \nabla \mathbf{W}_t^{m_2} \right)$ to $\tilde{\mathbf{W}} - \mathbf{P}' \tilde{\mathbf{W}} \mathbf{P}$.

**Assumption 7.** *Since $\|\eta (\nabla \mathbf{W}_t^{m_1})^\top \nabla \mathbf{W}_t^{m_2}\| \le \eta C \to 0$ when $\eta \to 0$, where $C$ denotes the product of the maximum gradient norms, we omit this high-order term and analyze the remaining terms. Therefore, our goal is to update the gradients through $\Delta \mathbf{W}_t^{m_1} := \tau_1(\nabla \mathbf{W}_t^{m_1})$ and $\Delta \mathbf{W}_t^{m_2} := \tau_2(\nabla \mathbf{W}_t^{m_1})$, and achieve $\left( (\mathbf{W}_{t-1}^{m_1})^\top \Delta \mathbf{W}_t^{m_2} + (\Delta \mathbf{W}_t^{m_1})^\top \mathbf{W}_{t-1}^{m_2} \right)$ closed to $\tilde{\mathbf{W}} - \mathbf{P}' \tilde{\mathbf{W}} \mathbf{P}$. $\tau_1$ and $\tau_2$ are the transformation functions for gradients.*

According to Lemma 9, we have the general solution for solving for $\mathbf{AX} + \mathbf{YB} = \mathbf{C}$:

$$\begin{cases} \mathbf{X} = \mathbf{A}^\dagger \mathbf{C} + \mathbf{A}^\dagger \mathbf{NB} + (\mathbf{I} - \mathbf{A}^\dagger \mathbf{A})\mathbf{M}, \\ \mathbf{Y} = (\mathbf{I} - \mathbf{AA}^\dagger)\mathbf{CB}^\dagger - \mathbf{N} + (\mathbf{I} - \mathbf{AA}^\dagger)\mathbf{NBB}^\dagger. \end{cases} \quad (37)$$

For simplification, let $\mathbf{A} = (\mathbf{W}_{t-1}^{m_1})^\top, \mathbf{B} = \mathbf{W}_{t-1}^{m_2}, \mathbf{X}_0 = \nabla \mathbf{W}_t^{m_2}, \mathbf{Y}_0 = (\nabla \mathbf{W}_t^{m_1})^\top$, and the right-hand side $\mathbf{C}$ being $\tilde{\mathbf{W}} - \mathbf{P}' \tilde{\mathbf{W}} \mathbf{P}$[18].

Before obtaining the solution, we consider whether $\mathbf{C}$ satisfies the condition. The first term $\tilde{\mathbf{W}} := \mathbf{AX}_0 + \mathbf{Y}_0 \mathbf{B}$ satisfies $(\mathbf{I} - \mathbf{AA}^\dagger)\tilde{\mathbf{W}}(\mathbf{I} - \mathbf{B}^\dagger \mathbf{B}) = \mathbf{0}$, as:

$$(\mathbf{I} - \mathbf{AA}^\dagger)(\mathbf{AA}_0 + \mathbf{Y}_0 \mathbf{B})(\mathbf{I} - \mathbf{B}^\dagger \mathbf{B}) = \underbrace{(\mathbf{I} - \mathbf{AA}^\dagger)\mathbf{A}}_{\mathbf{A} - \mathbf{A} = \mathbf{0}} \mathbf{X}(\mathbf{I} - \mathbf{B}^\dagger \mathbf{B}) + (\mathbf{I} - \mathbf{AA}^\dagger)\mathbf{Y}\underbrace{\mathbf{B}(\mathbf{I} - \mathbf{B}^\dagger \mathbf{B})}_{\mathbf{B} - \mathbf{B} = \mathbf{0}} = \mathbf{0}.$$

$$(38)$$

We then analyze the second term:

$$(\mathbf{I} - \mathbf{AA}^\dagger)\mathbf{P}' \tilde{\mathbf{W}} \mathbf{P}(\mathbf{I} - \mathbf{B}^\dagger \mathbf{B}) = (\mathbf{I} - \mathbf{AA}^\dagger)\mathbf{P}' \mathbf{AX}(\mathbf{I} - \mathbf{B}^\dagger \mathbf{B}) + (\mathbf{I} - \mathbf{AA}^\dagger)\mathbf{YBP}(\mathbf{I} - \mathbf{B}^\dagger \mathbf{B})$$

$$(39)$$

Suppose to satisfy $(\mathbf{I} - \mathbf{AA}^\dagger)\mathbf{P}' \tilde{\mathbf{W}} \mathbf{P}(\mathbf{I} - \mathbf{B}^\dagger \mathbf{B}) = \mathbf{0}$, the columns space of $\mathbf{Z}_{t-1}^{m_1}$ should be a subset of the columns space of the parameter $\mathbf{W}_{t-1}^{m_1}$ (*i.e.*, $\mathbf{A}$). Similarly, $\mathrm{col}(\mathbf{Z}_{t-1}^{m_2}) \subseteq \mathrm{col}(\mathbf{W}_{t-1}^{m_2})$. That means, at step $t - 1$, $\mathbf{W}_{t-1}$ should be trained well to capture the key features from the input data $\mathbf{Z}_{t-1}$. According to the Lemmas 10 and 11, the condition satisfies for the general solution.

Afterward, we can update the solutions:

$$\mathbf{Y} = (\mathbf{I} - \mathbf{AA}^\dagger)\mathbf{CB}^\dagger - \mathbf{N} + (\mathbf{I} - \mathbf{AA}^\dagger)\mathbf{NBB}^\dagger \quad (40)$$

---

[18] Here we omit the high-order term for simplicity in $\tilde{\mathbf{W}}$.

$$= (\mathbf{I} - \mathbf{A}\mathbf{A}^\dagger)(\tilde{\mathbf{W}} - \mathbf{P}'\tilde{\mathbf{W}}\mathbf{P})\mathbf{B}^\dagger - \mathbf{N} + (\mathbf{I} - \mathbf{A}\mathbf{A}^\dagger)\mathbf{N}\mathbf{B}\mathbf{B}^\dagger \tag{41}$$

$$= (\mathbf{I} - \mathbf{A}\mathbf{A}^\dagger)\tilde{\mathbf{W}}\mathbf{B}^\dagger - (\mathbf{I} - \mathbf{A}\mathbf{A}^\dagger)\mathbf{P}'\tilde{\mathbf{W}}\mathbf{P}\mathbf{B}^\dagger - \mathbf{N} + (\mathbf{I} - \mathbf{A}\mathbf{A}^\dagger)\mathbf{N}\mathbf{B}\mathbf{B}^\dagger \tag{42}$$

$$= (\mathbf{I} - \mathbf{A}\mathbf{A}^\dagger)(\mathbf{A}\mathbf{X}_0 + \mathbf{Y}_0\mathbf{B})\mathbf{B}^\dagger - (\mathbf{I} - \mathbf{A}\mathbf{A}^\dagger)\mathbf{P}'\tilde{\mathbf{W}}\mathbf{P}\mathbf{B}^\dagger - \mathbf{N} + (\mathbf{I} - \mathbf{A}\mathbf{A}^\dagger)\mathbf{N}\mathbf{B}\mathbf{B}^\dagger \quad (\text{replace } \tilde{\mathbf{W}}) \tag{43}$$

$$= (\mathbf{I} - \mathbf{A}\mathbf{A}^\dagger)\mathbf{A}\mathbf{X}_0\mathbf{B}^\dagger + (\mathbf{I} - \mathbf{A}\mathbf{A}^\dagger)\mathbf{Y}_0\mathbf{B}\mathbf{B}^\dagger - (\mathbf{I} - \mathbf{A}\mathbf{A}^\dagger)\mathbf{P}'\tilde{\mathbf{W}}\mathbf{P}\mathbf{B}^\dagger - \mathbf{N} + (\mathbf{I} - \mathbf{A}\mathbf{A}^\dagger)\mathbf{N}\mathbf{B}\mathbf{B}^\dagger \tag{44}$$

$$= (\mathbf{I} - \mathbf{A}\mathbf{A}^\dagger)\mathbf{A}\mathbf{X}_0\mathbf{B}^\dagger + (\mathbf{I} - \mathbf{A}\mathbf{A}^\dagger)\mathbf{Y}_0\mathbf{B}\mathbf{B}^\dagger - (\mathbf{I} - \mathbf{A}\mathbf{A}^\dagger)\mathbf{P}'\tilde{\mathbf{W}}\mathbf{P}\mathbf{B}^\dagger \tag{45}$$
$$+ \mathbf{Y}_0 + \mathbf{P}'\mathbf{Y}_0\mathbf{B}\mathbf{P}\mathbf{B}^\dagger + (\mathbf{I} - \mathbf{A}\mathbf{A}^\dagger)(-\mathbf{Y}_0 - \mathbf{P}'\mathbf{Y}_0\underbrace{\mathbf{B}\mathbf{P}\mathbf{B}^\dagger}_{\tilde{\mathbf{P}}})\mathbf{B}\mathbf{B}^\dagger \quad (\text{set } \mathbf{N} = -\mathbf{Y}_0 - \mathbf{P}'\mathbf{Y}_0\mathbf{B}\mathbf{P}\mathbf{B}^\dagger)$$

$$\tag{46}$$

$$= \mathbf{Y}_0 - \mathbf{P}'\mathbf{Y}_0\tilde{\mathbf{P}} = (\nabla\mathbf{W}_t^{m_1})^\top - \mathbf{P}'(\nabla\mathbf{W}_t^{m_1})^\top\tilde{\mathbf{P}}. \tag{47}$$

$$\mathbf{X} = \mathbf{A}^\dagger\mathbf{C} + \mathbf{A}^\dagger\mathbf{N}\mathbf{B} + (\mathbf{I} - \mathbf{A}^\dagger\mathbf{A})\mathbf{M} \tag{48}$$

$$= \mathbf{A}^\dagger(\tilde{\mathbf{W}} - \mathbf{P}'\tilde{\mathbf{W}}\mathbf{P}) + \mathbf{A}^\dagger\mathbf{N}\mathbf{B} + (\mathbf{I} - \mathbf{A}^\dagger\mathbf{A})\mathbf{M} \tag{49}$$

$$= \mathbf{A}^\dagger\tilde{\mathbf{W}} - \mathbf{A}^\dagger\mathbf{P}'\tilde{\mathbf{W}}\mathbf{P} + \mathbf{A}^\dagger\mathbf{N}\mathbf{B} + (\mathbf{I} - \mathbf{A}^\dagger\mathbf{A})\mathbf{M} \tag{50}$$

$$= \mathbf{A}^\dagger(\mathbf{A}\mathbf{X}_0 + \mathbf{Y}_0\mathbf{B}) - \mathbf{A}^\dagger\mathbf{P}'\tilde{\mathbf{W}}\mathbf{P} + \mathbf{A}^\dagger\mathbf{N}\mathbf{B} + (\mathbf{I} - \mathbf{A}^\dagger\mathbf{A})\mathbf{M} \tag{51}$$

$$= \mathbf{A}^\dagger\mathbf{A}\mathbf{X}_0 + \mathbf{A}^\dagger\mathbf{Y}_0\mathbf{B} - \mathbf{A}^\dagger\mathbf{P}'\tilde{\mathbf{W}}\mathbf{P} + \mathbf{A}^\dagger\mathbf{N}\mathbf{B} + (\mathbf{I} - \mathbf{A}^\dagger\mathbf{A})\mathbf{M} \tag{52}$$

$$= \mathbf{A}^\dagger\mathbf{A}\mathbf{X}_0 + \mathbf{A}^\dagger\mathbf{Y}_0\mathbf{B} - \mathbf{A}^\dagger\mathbf{P}'\tilde{\mathbf{W}}\mathbf{P} - \mathbf{A}^\dagger\mathbf{Y}_0\mathbf{B} - \mathbf{A}^\dagger\mathbf{P}'\mathbf{Y}_0\mathbf{B}\mathbf{P}\mathbf{B}^\dagger\mathbf{B} + (\mathbf{I} - \mathbf{A}^\dagger\mathbf{A})\mathbf{X}_0 \quad (\text{set } \mathbf{M} = \mathbf{X}_0) \tag{53}$$

$$= \mathbf{X}_0 - \tilde{\mathbf{P}}'\mathbf{X}_0\mathbf{P} = \nabla\mathbf{W}_t^{m_2} - \tilde{\mathbf{P}}'\nabla\mathbf{W}_t^{m_2}\mathbf{P}. \tag{54}$$

Therefore, we can update the gradients:

$$\begin{cases} \Delta\mathbf{W}_t^{m_1} := \nabla\mathbf{W}_t^{m_1} - \tilde{\mathbf{P}}\nabla\mathbf{W}_t^{m_1}\mathbf{P}', \\ \Delta\mathbf{W}_t^{m_2} := \nabla\mathbf{W}_t^{m_2} - \tilde{\mathbf{P}}'\nabla\mathbf{W}_t^{m_2}\mathbf{P}, \end{cases} \tag{55}$$

$$\square$$

## B.3 The Upper Bound of Loss of Stability

> **Recall:**
>
> **Theorem 5.** *In CMCL, the upper bound of loss of stability is:*
> $$\|\mathbf{A}_{t-1;t}^{m_1,m_2} - \mathbf{A}_{t-1;t-1}^{m_1,m_2}\|_2 \le \eta^2 \cdot \|\mathbf{Z}_{t-1;*}^{m_1}\|_2\|\mathbf{Z}_{t-1;*}^{m_2}\|_2 \cdot \mathcal{F}(\nabla\mathbf{W}), \tag{56}$$
> *where $\mathcal{F}$ represents the interactions between different updated gradients $\nabla\mathbf{W}_t^{m_1}$ and $\nabla\mathbf{W}_t^{m_2}$. Specifically, $\mathcal{F}(\nabla\mathbf{W}) := 2\|\nabla\mathbf{W}_t^{m_1}\|_2\|\nabla\mathbf{W}_t^{m_2}\|_2 + \|\nabla\mathbf{W}_t^{m_1}\|_2^2 + \|\nabla\mathbf{W}_t^{m_2}\|_2^2$.*

*Proof.* Here we define the measuring of the loss of stability, which is implemented by the difference between $\mathbf{A}_{t-1;t}^{m_1,m_2}$ and $\mathbf{A}_{t-1;t-1}^{m_1,m_2}$.

$$\|\mathbf{A}_{t-1;t}^{m_1,m_2} - \mathbf{A}_{t-1;t-1}^{m_1,m_2}\|_2 = \|\eta(\mathbf{Z}_{t-1;*}^{m_1})^\top \left((\mathbf{W}_{t-1}^{m_1})^\top\Delta\mathbf{W}_t^{m_2} + (\Delta\mathbf{W}_t^{m_1})^\top\mathbf{W}_{t-1}^{m_2} - \eta(\Delta\mathbf{W}_t^{m_1})^\top\Delta\mathbf{W}_t^{m_2}\right)\mathbf{Z}_{t-1;*}^{m_2}\|_2. \tag{57}$$

We first analyze each term:

$$(\mathbf{W}_{t-1}^{m_1})^\top\Delta\mathbf{W}_t^{m_2} = (\mathbf{W}_{t-1}^{m_1})^\top\nabla\mathbf{W}_t^{m_2} - (\mathbf{W}_{t-1}^{m_1})^\top\tilde{\mathbf{P}}'\nabla\mathbf{W}_t^{m_2}\mathbf{P}. \tag{58}$$

$$(\Delta\mathbf{W}_t^{m_1})^\top\mathbf{W}_{t-1}^{m_2} = (\nabla\mathbf{W}_t^{m_1})^\top\mathbf{W}_{t-1}^{m_2} - \mathbf{P}'(\nabla\mathbf{W}_t^{m_1})^\top\tilde{\mathbf{P}}\mathbf{W}_{t-1}^{m_2}. \tag{59}$$

$$(\Delta \mathbf{W}_t^{m_1})^\top)\Delta \mathbf{W}_t^{m_2} = \left(\nabla \mathbf{W}_t^{m_1} - \tilde{\mathbf{P}}\nabla \mathbf{W}_t^{m_1}\mathbf{P}'\right)^\top \left(\nabla \mathbf{W}_t^{m_2} - \tilde{\mathbf{P}}'\nabla \mathbf{W}_t^{m_2}\mathbf{P}\right) \tag{60}$$

$$= (\nabla \mathbf{W}_t^{m_1})^\top \nabla \mathbf{W}_t^{m_2} - (\nabla \mathbf{W}_t^{m_1})^\top \tilde{\mathbf{P}}'\nabla \mathbf{W}_t^{m_2}\mathbf{P} \tag{61}$$

$$- \mathbf{P}'(\nabla \mathbf{W}_t^{m_1})^\top \tilde{\mathbf{P}}\nabla \mathbf{W}_t^{m_2} + \mathbf{P}'(\nabla \mathbf{W}_t^{m_1})^\top \tilde{\mathbf{P}}\tilde{\mathbf{P}}'\nabla \mathbf{W}_t^{m_2}\mathbf{P}. \tag{62}$$

Thus, we update the deviation computation:

$$\|\mathbf{A}_{t-1;t}^{m_1,m_2} - \mathbf{A}_{t-1;t-1}^{m_1,m_2}\|_2 \tag{63}$$

$$= \|\eta \underbrace{\mathbf{Z}_{t-1;*}^{m_1}(\tilde{\mathbf{W}} - \mathbf{P}'\tilde{\mathbf{W}}\mathbf{P})\mathbf{Z}_{t-1;*}^{m_2}}_{=\mathbf{0}} - \eta\mathbf{Z}_{t-1;*}^{m_1}(\eta(\Delta \mathbf{W}_t^{m_1})^\top \Delta \mathbf{W}_t^{m_2})\mathbf{Z}_{t-1;*}^{m_2}\|_2 \tag{64}$$

$$= \|\eta\mathbf{Z}_{t-1;*}^{m_1}(\eta(\Delta \mathbf{W}_t^{m_1})^\top \Delta \mathbf{W}_t^{m_2})\mathbf{Z}_{t-1;*}^{m_2}\|_2 \tag{65}$$

$$\leq \eta^2\|\mathbf{Z}_{t-1;*}^{m_1}\|_2\|\mathbf{Z}_{t-1;*}^{m_2}\|_2 \qquad \text{(sub-multiplicative property of matrix norms)} \tag{66}$$

$$\left(\|(\nabla \mathbf{W}_t^{m_1})^\top \nabla \mathbf{W}_t^{m_2})\|_2 + \|(\nabla \mathbf{W}_t^{m_1})^\top \tilde{\mathbf{P}}'\nabla \mathbf{W}_t^{m_2}\mathbf{P}\|_2 \right. \tag{67}$$

$$\left. + \|\mathbf{P}'(\nabla \mathbf{W}_t^{m_1})^\top \tilde{\mathbf{P}}\nabla \mathbf{W}_t^{m_2})\|_2 + \|\mathbf{P}'(\nabla \mathbf{W}_t^{m_1})^\top \tilde{\mathbf{P}}\tilde{\mathbf{P}}'\nabla \mathbf{W}_t^{m_2}\mathbf{P}\|_2\right) \qquad \text{(triangle inequality)} \tag{68}$$

$$\leq \eta^2\|\mathbf{Z}_{t-1;*}^{m_1}\|_2\|\mathbf{Z}_{t-1;*}^{m_2}\|_2 \tag{69}$$

$$\left(\|\nabla \mathbf{W}_t^{m_1}\|_2\|\nabla \mathbf{W}_t^{m_2}\|_2 + \|\nabla \mathbf{W}_t^{m_1}\|_2^2 + \|\nabla \mathbf{W}_t^{m_2}\|_2^2 + \|\nabla \mathbf{W}_t^{m_1}\|_2\|\nabla \mathbf{W}_t^{m_2}\|_2\right). \qquad (\|\mathbf{P}\|_2 \leq 1) \tag{70}$$

Here we use $\mathcal{F}$ to denote the interactions among different gradient norms $\|\nabla \mathbf{W}_t\|$:

$$\mathcal{F}(\nabla \mathbf{W}) := 2\|\nabla \mathbf{W}_t^{m_1}\|_2\|\nabla \mathbf{W}_t^{m_2}\|_2 + \|\nabla \mathbf{W}_t^{m_1}\|_2^2 + \|\nabla \mathbf{W}_t^{m_2}\|_2^2. \tag{71}$$

Therefore, we obtain the upper bound of loss of stability:

$$\|\mathbf{A}_{t-1;t}^{m_1,m_2} - \mathbf{A}_{t-1;t-1}^{m_1,m_2}\|_2 \leq \eta^2 \cdot \|\mathbf{Z}_{t-1;*}^{m_1}\|_2\|\mathbf{Z}_{t-1;*}^{m_2}\|_2 \cdot \mathcal{F}(\nabla \mathbf{W}). \tag{72}$$

$$\square$$

## B.4  The Upper Bound of Plasticity

> **Recall:**
>
> **Theorem 6.** *By transforming the gradients, the loss $\mathcal{L}_t$ update at step $t$ from the previous loss $\mathcal{L}_{t-1}$ is following:*
>
> $$\mathcal{L}_t - \mathcal{L}_{t-1} \leq 0 \tag{73}$$
>
> *to keep the plasticity when $\frac{o(\eta)}{\eta} \leq 0$ at the time $t$.*

*Proof.* Here we consider the capability to learn from new modality paired data (plasticity), and provide the upper bound from the loss optimization perspective. We know that

$$\underbrace{\mathcal{L}(\mathbf{W}_t^{m_1}, \mathbf{W}_t^{m_2})}_{\mathcal{L}_t} = \underbrace{\mathcal{L}(\mathbf{W}_{t-1}^{m_1}, \mathbf{W}_{t-1}^{m_2})}_{\mathcal{L}_{t-1}} - \eta\left(\langle\nabla \mathbf{W}_t^{m_1}, \Delta \mathbf{W}_t^{m_1}\rangle + \langle\nabla \mathbf{W}_t^{m_2}, \Delta \mathbf{W}_t^{m_2}\rangle\right) + o(\eta). \tag{74}$$

Analyze the first-order term:

$$\langle\nabla \mathbf{W}_t^{m_1}, \Delta \mathbf{W}_t^{m_1}\rangle + \langle\nabla \mathbf{W}_t^{m_2}, \Delta \mathbf{W}_t^{m_2}\rangle \tag{75}$$

$$= \langle \nabla \mathbf{W}_t^{m_1}, \nabla \mathbf{W}_t^{m_1} - \tilde{\mathbf{P}} \nabla \mathbf{W}_t^{m_1} \mathbf{P}' \rangle + \langle \nabla \mathbf{W}_t^{m_2}, \nabla \mathbf{W}_t^{m_2} - \tilde{\mathbf{P}}' \nabla \mathbf{W}_t^{m_2} \mathbf{P} \rangle \tag{76}$$

$$= \|\nabla \mathbf{W}_t^{m_1}\|_F^2 + \|\nabla \mathbf{W}_t^{m_2}\|_F^2 - \langle \nabla \mathbf{W}_t^{m_1}, \tilde{\mathbf{P}} \nabla \mathbf{W}_t^{m_1} \mathbf{P}' \rangle - \langle \nabla \mathbf{W}_t^{m_2}, \tilde{\mathbf{P}}' \nabla \mathbf{W}_t^{m_2} \mathbf{P} \rangle \tag{77}$$

$$= \|\nabla \mathbf{W}_t^{m_1}\|_F^2 - \langle \nabla \mathbf{W}_t^{m_1}, \tilde{\mathbf{P}} \nabla \mathbf{W}_t^{m_1} \mathbf{P}' \rangle + \|\nabla \mathbf{W}_t^{m_2}\|_F^2 - \langle \nabla \mathbf{W}_t^{m_2}, \tilde{\mathbf{P}}' \nabla \mathbf{W}_t^{m_2} \mathbf{P} \rangle \tag{78}$$

$$= \|\nabla \mathbf{W}_t^{m_1}\|_F^2 - \langle \nabla \mathbf{W}_t^{m_1}, \tilde{\mathbf{P}} \nabla \mathbf{W}_t^{m_1} \mathbf{P}' \rangle + \|\nabla \mathbf{W}_t^{m_2}\|_F^2 - \langle \nabla \mathbf{W}_t^{m_2}, \tilde{\mathbf{P}}' \nabla \mathbf{W}_t^{m_2} \mathbf{P} \rangle \tag{79}$$

$$\geq \|\nabla \mathbf{W}_t^{m_1}\|_F^2 - \langle \nabla \mathbf{W}_t^{m_1}, \nabla \mathbf{W}_t^{m_1} \rangle + \|\nabla \mathbf{W}_t^{m_2}\|_F^2 - \langle \nabla \mathbf{W}_t^{m_2}, \nabla \mathbf{W}_t^{m_2} \rangle \tag{80}$$

$$= \|\nabla \mathbf{W}_t^{m_1}\|_F^2 - \|\nabla \mathbf{W}_t^{m_1}\|_F^2 + \|\nabla \mathbf{W}_t^{m_2}\|_F^2 - \|\nabla \mathbf{W}_t^{m_2}\|_F^2 \tag{81}$$

$$= 0 \tag{82}$$

Since $\frac{|o(\eta)|}{\eta} \to 0$ when $\eta \to 0$, there exists a $\bar{\eta}$ such that $|o(\eta)| < \eta \Big( \langle \nabla \mathbf{W}_t^{m_1}, \Delta \mathbf{W}_t^{m_1} \rangle + \langle \nabla \mathbf{W}_t^{m_2}, \Delta \mathbf{W}_t^{m_2} \rangle \Big)$. Therefore, $\mathcal{L}_t - \mathcal{L}_{t-1} \leq 0$. $\qquad \square$

### B.5 Extension to Any Steps

**Theorem 8.** *Let $\bar{\mathbf{Z}}_{<t} = \bar{\mathbf{Z}}_{<t-1} + \tilde{\mathbf{Z}}_{t-1}$, where $\tilde{\mathbf{Z}}_{t-1} = \frac{1}{n_{t-1}} \mathbf{Z}_{t-1} \mathbf{Z}_{t-1}^\top$. Then, the column space of $\bar{\mathbf{Z}}_{<t}$ is equal to the column space of $[\mathbf{Z}_1, \mathbf{Z}_2, \ldots, \mathbf{Z}_{t-1}]$.*

*Proof.* First consider the base case ($t = 2$). The column space of $\bar{\mathbf{Z}}_{<2}$ is $\text{span}\{\mathbf{Z}_1\}$, which is equal to the column space of $[\mathbf{Z}_1]$. $\bar{\mathbf{Z}}_{<2} = \tilde{\mathbf{Z}}_1 = \frac{1}{n_1} \mathbf{Z}_1 \mathbf{Z}_1^\top$. Thus, the statement holds for $t = 2$.

Assume that for some $k \geq 2$, $\text{Col}(\bar{\mathbf{Z}}_{<k}) = \text{Col}([\mathbf{Z}_1, \mathbf{Z}_2, \ldots, \mathbf{Z}_{k-1}])$. Then, consider the inductive Step ($t = k + 1$). By definition, $\bar{\mathbf{Z}}_{<k+1} = \bar{\mathbf{Z}}_{<k} + \tilde{\mathbf{Z}}_k$. From the inductive hypothesis, $\text{Col}(\bar{\mathbf{Z}}_{<k}) = \text{span}\{\mathbf{Z}_1, \mathbf{Z}_2, \ldots, \mathbf{Z}_{k-1}\}$. Since $\tilde{\mathbf{Z}}_k = \frac{1}{n_k} \mathbf{Z}_k \mathbf{Z}_k^\top$, its column space is $\text{span}\{\mathbf{Z}_k\}$. The column space of $\bar{\mathbf{Z}}_{<k+1}$ is the sum of the column spaces of $\bar{\mathbf{Z}}_{<k}$ and $\tilde{\mathbf{Z}}_k$, which is:

$$\text{Col}(\bar{\mathbf{Z}}_{<k+1}) = \text{Col}(\bar{\mathbf{Z}}_{<k}) + \text{Col}(\tilde{\mathbf{Z}}_k) = \text{span}\{\mathbf{Z}_1, \mathbf{Z}_2, \ldots, \mathbf{Z}_{k-1}\} + \text{span}\{\mathbf{Z}_k\} = \text{span}\{\mathbf{Z}_1, \mathbf{Z}_2, \ldots, \mathbf{Z}_k\} \tag{83}$$

The column space of $[\mathbf{Z}_1, \mathbf{Z}_2, \ldots, \mathbf{Z}_k]$ is also $\text{span}\{\mathbf{Z}_1, \mathbf{Z}_2, \ldots, \mathbf{Z}_k\}$. Therefore, $\text{Col}(\bar{\mathbf{Z}}_{<k+1}) = \text{Col}([\mathbf{Z}_1, \mathbf{Z}_2, \ldots, \mathbf{Z}_k])$, and the statement holds for $t = k + 1$. $\qquad \square$

Due to the same column space, we can use the feature covariance to compute the projection matrix to save the computational cost.

## C Technical Lemmas

**Lemma 1** (Four properties of pseudo-inverse matrix). *If $\mathbf{A}^\dagger$ is the pseudo-inverse matrix of $\mathbf{A}$, then it satisfies:*

$$\mathbf{A} \mathbf{A}^\dagger \mathbf{A} = \mathbf{A}, \quad \mathbf{A}^\dagger \mathbf{A} \mathbf{A}^\dagger = \mathbf{A}^\dagger, \quad (\mathbf{A} \mathbf{A}^\dagger)^\top = \mathbf{A} \mathbf{A}^\dagger, \quad (\mathbf{A}^\dagger \mathbf{A})^\top = \mathbf{A}^\dagger \mathbf{A}. \tag{84}$$

**Lemma 2** (Idempotent property). *For the matrix $\mathbf{P} = \mathbf{I} - \mathbf{U}\mathbf{U}^\top$, where $\{\mathbf{U}, \boldsymbol{\Lambda}, \mathbf{U}\} = SVD(\mathbf{A})$, it satisfies $\mathbf{P}^2 = \mathbf{P}$.*

*Proof.*

$$\mathbf{P}^2 = (\mathbf{I} - \mathbf{U}\mathbf{U}^\top)(\mathbf{I} - \mathbf{U}\mathbf{U}^\top) = \mathbf{I} + \underbrace{\mathbf{U}\mathbf{U}^\top\mathbf{U}\mathbf{U}^\top}_{\mathbf{U}\mathbf{U}^\top} - \mathbf{U}\mathbf{U}^\top - \mathbf{U}\mathbf{U}^\top = \mathbf{I} - \mathbf{U}\mathbf{U}^\top = \mathbf{P} \tag{85}$$

$\qquad \square$

**Lemma 3** (Symmetry property). *For the matrix $\mathbf{P} = \mathbf{I} - \mathbf{U}\mathbf{U}^\top$, where $\{\mathbf{U}, \boldsymbol{\Lambda}, \mathbf{U}\} = SVD(\mathbf{A})$, it satisfies $\mathbf{P}^\top = \mathbf{P}$.*

*Proof.* $\mathbf{P}^\top = \mathbf{I}^\top - (\mathbf{U}\mathbf{U}^\top)^\top = \mathbf{I} - \mathbf{U}\mathbf{U}^\top = \mathbf{P}$. $\qquad\qquad\square$

**Lemma 4** (Eigenvalues of idempotent matrix). *For an idempotent matrix* $\mathbf{P}$*, satisfies* $\mathbf{P}^2 = \mathbf{P}$*, its eigenvalues* $\lambda \in \{0, 1\}$.

*Proof.* If $\mathbf{P}v = \lambda v$ for the nonzero vector $v$, then apply $\mathbf{P}$ again to both sides of the equation:

$$\mathbf{P}^2 v = \mathbf{P}(\mathbf{P}v) = \mathbf{P}(\lambda v) = \lambda \mathbf{P}v = \lambda^2 v \tag{86}$$

$$= \mathbf{P}v = \lambda v. \tag{87}$$

Thus, we have $\lambda^2 v = \lambda v$, which indicates $\lambda \in \{1, 0\}$ since $v \neq \mathbf{0}$. $\qquad\square$

**Lemma 5.** *A matrix* $\mathbf{A}$ *is positive semi-definite if and only if all its eigenvalues are non-negative.*

**Lemma 6.** *If the Hessian matrix* $\mathbf{H}$ *of* $f(\cdot)$ *is positive semidefinite (PSD) for all* $x$ *in the domain, then* $f(\cdot)$ *is a convex function.*

**Lemma 7** (Solving $\mathbf{AXB} = \mathbf{C}$ [57]). *A necessary and sufficient condition for the equation* $\mathbf{AXB} = \mathbf{C}$ *to have a solution is*

$$\mathbf{AA}^\dagger \mathbf{XB}^\dagger \mathbf{B} = \mathbf{C}, \tag{88}$$

*in which case the general solution is*

$$\mathbf{X} = \underbrace{\mathbf{A}^\dagger \mathbf{CB}^\dagger}_{particular} + \underbrace{\mathbf{Y} - \mathbf{A}^\dagger \mathbf{AYBB}^\dagger}_{homogeneous}, \tag{89}$$

*where* $\mathbf{Y}$ *is arbitrary.*

**Lemma 8** (Solving $\mathbf{AX} - \mathbf{YB} = \mathbf{C}$ [58]). *Let* $\mathbf{A} \in \mathbb{R}^{m \times k}$*,* $\mathbf{B} \in \mathbb{R}^{l \times n}$ *and* $\mathbf{C} \in \mathbb{R}^{m \times n}$*. For*

$$\mathbf{AX} - \mathbf{YB} = \mathbf{C}, \tag{90}$$

*the equation has a solution* $\mathbf{X} \in \mathbb{R}^{k \times n}, \mathbf{Y} \in \mathbb{R}^{m \times l}$ *if and only if*

$$(\mathbf{I} - \mathbf{AA}^\dagger)\mathbf{C}(\mathbf{I} - \mathbf{B}^\dagger \mathbf{B}) = \mathbf{0}. \tag{91}$$

*If this is the case, the general solution of (90) has the form*

$$\begin{cases} \mathbf{X} = \mathbf{A}^\dagger \mathbf{C} + \mathbf{A}^\dagger \mathbf{NB} + (\mathbf{I} - \mathbf{A}^\dagger \mathbf{A})\mathbf{M}, \\ \mathbf{Y} = -(\mathbf{I} - \mathbf{AA}^\dagger)\mathbf{CB}^\dagger + \mathbf{N} - (\mathbf{I} - \mathbf{AA}^\dagger)\mathbf{NBB}^\dagger. \end{cases} \tag{92}$$

*with* $\mathbf{M} \in \mathbb{R}^{k \times n}$ *and* $\mathbf{N} \in \mathbb{R}^{m \times l}$ *being arbitrary.*

**Lemma 9** (Solving $\mathbf{AX} + \mathbf{YB} = \mathbf{C}$). $\mathbf{AX} + \mathbf{YB} = \mathbf{C}$ *has a solution if and only if*

$$(\mathbf{I} - \mathbf{AA}^\dagger)\mathbf{C}(\mathbf{I} - \mathbf{B}^\dagger \mathbf{B}) = \mathbf{0}. \tag{93}$$

*If this is the case, the general solution is:*

$$\begin{cases} \mathbf{X} = \mathbf{A}^\dagger \mathbf{C} + \mathbf{A}^\dagger \mathbf{NB} + (\mathbf{I} - \mathbf{A}^\dagger \mathbf{A})\mathbf{M}, \\ \mathbf{Y} = (\mathbf{I} - \mathbf{AA}^\dagger)\mathbf{CB}^\dagger - \mathbf{N} + (\mathbf{I} - \mathbf{AA}^\dagger)\mathbf{NBB}^\dagger. \end{cases} \tag{94}$$

**Lemma 10.** *If* $(\mathbf{I} - \mathbf{AA}^\dagger)\mathbf{P}'\mathbf{A} = \mathbf{0}$ *where* $\mathbf{P}'$ *is a projection matrix, then* $\mathbf{AA}^\dagger \mathbf{P}'\mathbf{A} = \mathbf{P}'\mathbf{A}$.

**Lemma 11.** *If* $\mathbf{AA}^\dagger \mathbf{P}'\mathbf{A} = \mathbf{P}'\mathbf{A}$*, where* $\mathbf{P}$ *is an orthogonal projection matrix, then* $\mathbf{A}^\dagger \mathbf{P}'\mathbf{A}$ *is a projection matrix as well.*

*Proof.* If $\mathbf{P}'$ is an orthogonal projection matrix, it satisfies $\mathbf{P}'^2 = \mathbf{P}'$, since $(\mathbf{A}^\dagger \mathbf{P}'\mathbf{A})^2 = \mathbf{A}^\dagger \mathbf{P}'\mathbf{AA}^\dagger \mathbf{P}'\mathbf{A} = \mathbf{A}^\dagger \mathbf{P}'\mathbf{P}'\mathbf{A} = \mathbf{A}^\dagger \mathbf{P}'\mathbf{A}$, $\mathbf{A}^\dagger \mathbf{P}'\mathbf{A}$ is a projection matrix. $\qquad\square$

# D  Related Literature

We review the topics of multimodal contrastive learning and continual learning, which are highly related to this work.

## D.1 Multimodal Contrastive Learning

Multimodal learning endows the model with the capability to understand and generate multiple modalities [59, 60, 61, 62, 63, 64, 65]. Multimodal contrastive learning builds upon the principles of cross-modal contrastive learning and has demonstrated remarkable success across various tasks [66, 9, 67, 68]. The fundamental philosophy of contrastive learning is to align different views of the same instance while distinguishing between distinct instances. In unimodal contrastive learning, an instance is augmented into two different views and brought closer in the feature space, following the paradigm of self-supervised learning [6, 69]. This concept extends naturally to cross-modal scenarios, as exemplified by models such as CLIP [10], ALIGN [27], VideoCLIP [26], and CLAP [25]. Recently, there has been growing interest in multimodal contrastive learning, which incorporates a broader range of modalities, *e.g.*, AudioCLIP [70] and WAV2CLIP [71]. Besides, ImageBind [11] introduces a vision-centric approach by leveraging large-scale multimodal data. Similarly, LanguageBind [15] and PointBind [72] shift the focus toward text and point cloud modalities, respectively. Other methods either enhance these frameworks (*e.g.*, FreeBind [14]) or explore alternative center-free strategies (*e.g.*, OmniBind [73] and UniBind [13]).

Multimodal contrastive learning empowers models to learn rich and informative latent representations across different modalities. This enhances AI's ability to comprehend inter-modal relationships, which is necessitated for many real-world tasks (*e.g.*, multimodal search [74, 75], multimodal generation [76, 77, 78], and recent multimodal AI-assistant [79, 20, 12, 80]). Furthermore, the exploration of modality uniqueness and commonness (*i.e.*, invariance) contributes to advancing the foundational research on machine learning and deep learning [81, 82, 83].

Despite the significance, the innate data-intensiveness raises questions on how to learn it efficiently. Large-scale multimodal datasets are difficult to acquire in a single collection process, and training models on such diverse data from scratch incurs substantial computational costs. A more practical and scalable way is to incrementally integrate emerging data into existing modality-aligned models. Motivated by this, we systematically investigate this research problem and propose a novel method with rigorous theoretical analysis and guarantees.

## D.2 Continual Learning

Continual learning, also referred to as incremental learning, aims to enable models to learn from new data effectively in a streaming way while retaining previously acquired knowledge [84, 85, 86, 87]. The primary challenge lies in integrating new information without causing *catastrophic forgetting*. Among existing approaches, replay-based methods have showcased superior performance through a buffer memory [88]. This buffer typically stores the previous samples, which are then leveraged in the subsequent training steps under additional constraints (*e.g.*, meta-experience replay (MER) [89], gradient episodic memory (GEM) [38], and dark-experience replay (DER) [39]). Another prominent category, regularization-based methods, mainly focuses on stabilizing network updates. These methods either constrain the variance of network parameters (weight regularization [90, 40, 91, 92]) or regulate the model's output space (function regularization [93, 94]) to mitigate forgetting. In contrast, parameter isolation methods dynamically allocate distinct subsets of model parameters to different tasks, thereby preventing interference and reducing forgetting [95, 96, 97]. Existing continual learning works mainly focus on two settings of task-incremental learning (TIL) [98] and class-incremental learning (CIL) [99]. Recent advancements have extended continual learning to multimodal scenarios, giving rise to multimodal continual learning (MMCL) [2]. Beyond conventional CL methods, MMCL incorporates prompt-based approaches as a specialized adaptation [100, 101, 102]. These methods introduce learnable prompt parameters as part of the input, enabling the model to differentiate tasks based on the learned prompts.

Research in continual learning has advanced significantly while leaving a blank in continual multimodal contrastive learning (CMCL). Some solutions from continual uni-modal or cross-modal contrastive learning show potential while overlooking the modality complexity. Existing MMCL approaches primarily focus on task-specific continual learning scenarios, neglecting fundamental representation-level challenges, which are task-agnostic. The absence of task boundaries and introduced modality complexity make extending these methods to CMCL highly challenging. To fill this blank, we take the initiative to investigate CMCL and propose solutions with theoretical analyses.

We believe our contributions provide a strong foundation for future research, which inspires further advancements in both academia and industry.

# E    Supplementary Experimental Settings

In this section, we elaborate on the experimental settings, especially concerning the baseline implementation. For all buffer-required baselines, like GEM and DER, we set the number of buffer memory to 256. In the following, we list the specialized settings for each baseline. We implement the *Vanilla* training with the same settings with DNS, while omitting the specialized one of $\lambda_{\min}$. *GEM* is equipped with a margin of 0.5. For *DER*, we set the penalty weight to 0.1. Note that *DER++* is built upon DER. The additional hyperparameter of *DER++*, *i.e.*, $\alpha$, which controls the strength of ground-truth prediction, is set to 0.1 for the case where LanguageBind is the backbone and 0.001 for ImageBind and UniBind. For *EWC*, the regulation for the quadratic constraint term is set to 0.5. For $Co^2L$, the distillation power is set to 1.0 and the temperature in CL is set to 1. *C-FLAT* uses AdamW as the base optimizer which is the default optimizer in our evaluation, and the gradient reduction method is the mean. The radius $\rho$ is 0.2 for the gradient norm, and the balance hyperparameter of loss terms is also set to 0.2. For *CILA*, the balancing distillation coefficient is set to 5.0. All the experiments are conducted with 4×NVIDIA RTX A5000 GPUs and 256GB memory.

# F    Additional Experimental Results

We report the detailed experimental results in this section.

## F.1    hyperparameter Analysis

We provide the training loss and performance (recall@5 and accuracy) changes in terms of different hyperparameter settings, including weight decays and minimum eigenvalues in truncated SVD. The results are illustrated in Figures 7, 8, and 9, corresponding to different modality-binding methods. It demonstrates that our method, DNS, can achieve superior performance compared to vanilla training and maintain the performance over time.

## F.2    Performance variations across Steps

We also track the training loss and recall variations at each step, as illustrated in Figures 10, 11, and 12. In most cases, DNS exhibits a decreasing training loss and improving performance, particularly when using ImageBind and UniBind as backbones. Moreover, DNS maintains stable training loss and performance after completing its designated training step.

# G    Broader Impact Statement

This research aims to contribute positively to the machine learning and multimodal learning fields by advancing continual multimodal contrastive learning, which enables models to dynamically integrate and retain knowledge from diverse multimodal data sources over time. Although we believe our work is unlikely to have direct negative societal impacts, we acknowledge the importance of considering potential misuse scenarios, such as the exploitation of continuously adapting systems in unethical surveillance or misinformation campaigns. The broader implication of our study is that it elevates multimodal models with the capability to seamlessly adapt to evolving multimodal environments with minimal retraining, which makes them particularly suitable for real-time applications in adaptive human-computer interaction and autonomous systems. Such advancements could lead to more versatile, resilient, and context-aware AI applications across various real-world domains.

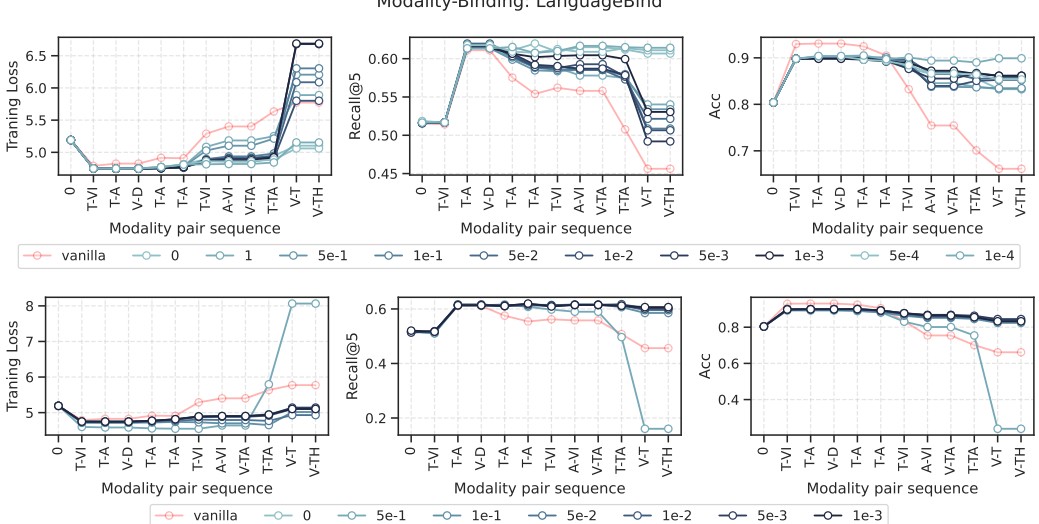

Figure 8: Detailed training loss (*left*), recall@5 (*middle*), and accuracy (*right*) results on the data at the first step. The top figure indicates the results about different $\lambda_{\min}$, while the bottom is for different weight decays, with *LanguageBind* as the backbone. The red line indicates the performance of the vanilla method in CMCL.

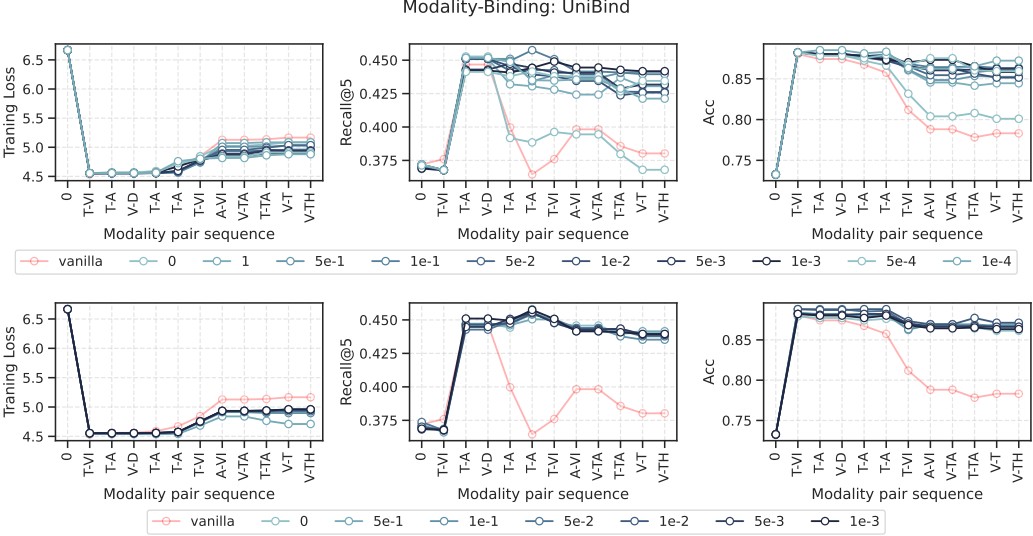

Figure 9: Detailed training loss (*left*), recall@5 (*middle*), and accuracy (*right*) results on the data at the first step. The top figure indicates the results about different $\lambda_{\min}$, while the bottom is for different weight decays, with *UniBind* as the backbone. The red line indicates the performance of the vanilla method in CMCL.

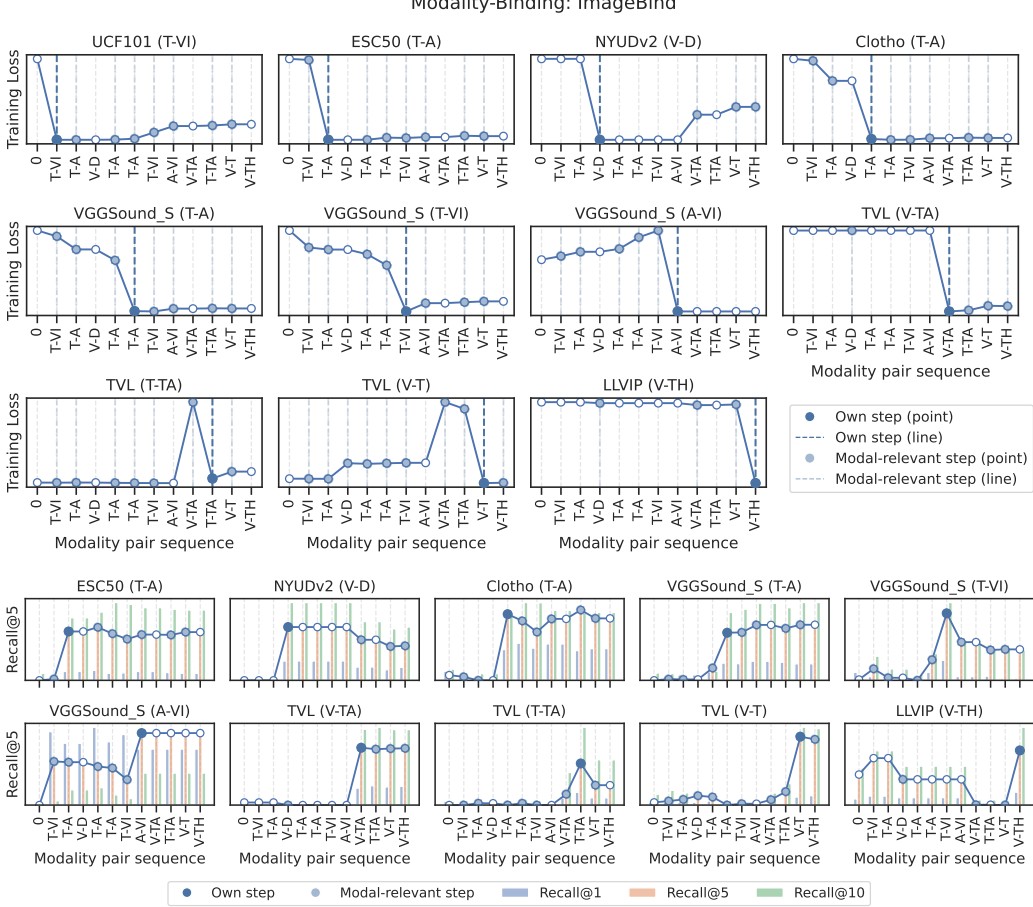

Figure 10: Detailed experimental results of the training loss (*the top three rows*) and recall (*the last two rows*) over a sequence of modality pair data, with *ImageBind* as the modality-binding backbone. Notably, the y-values are normalized for better visualization.

# H   Reproducibility

We provide comprehensive implementation details, including illustrative algorithm flows and core pseudo-code. Additionally, source codes are publicly released.

# I   Limitations

This paper fills in the blank of continual multimodal contrastive learning in the development of both multimodal learning and continual learning research. Specifically, this paper introduces a method that projects the gradient onto specialized subspaces, where the incorporation of new data does not interfere with previously acquired knowledge. Although our work is supported by systematic theoretical analyses and empirical studies, it does not yet examine the generalization errors [103] of the proposed method in downstream tasks. We regard addressing the limitation as our future direction.

# J   Potential in Scientific Fields

The CMCL framework holds significant potential for applications in complex fields like medicine and neuroscience, where modalities such as MRI, CT scans, and electrophysiological signals re-

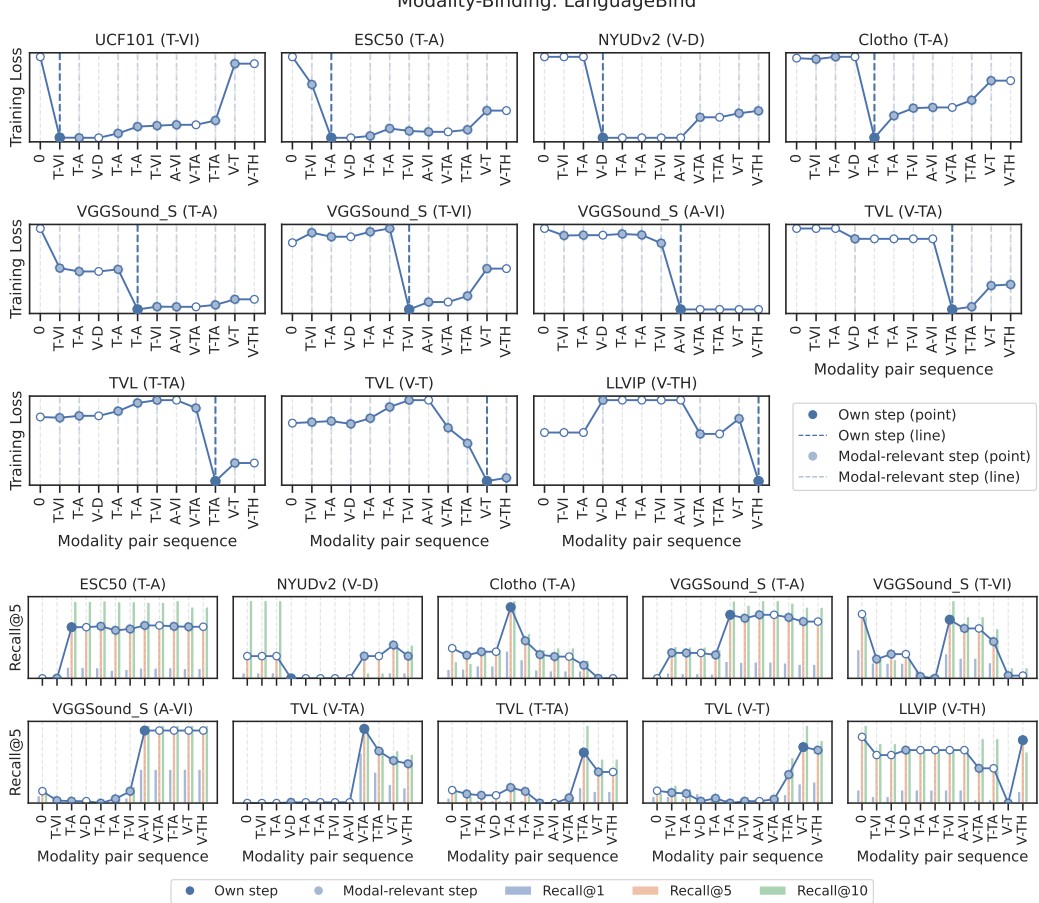

Figure 11: Detailed experimental results of the training loss (*the top three rows*) and recall (*the last two rows*) over a sequence of modality pair data, with *LanguageBind* as the modality-binding backbone. Notably, the y-values are normalized for better visualization.

quire incremental integration. In medical diagnostics, CMCL can incrementally incorporate diverse imaging modalities. In neuroscience research, our framework can facilitate the progressive understanding of neural correlates by continually integrating modalities like functional MRI, EEG, and MEG, thereby providing richer insights into brain functions and disorders. CMCL presents an exciting avenue for future interdisciplinary research to enhance both interpretability and performance.

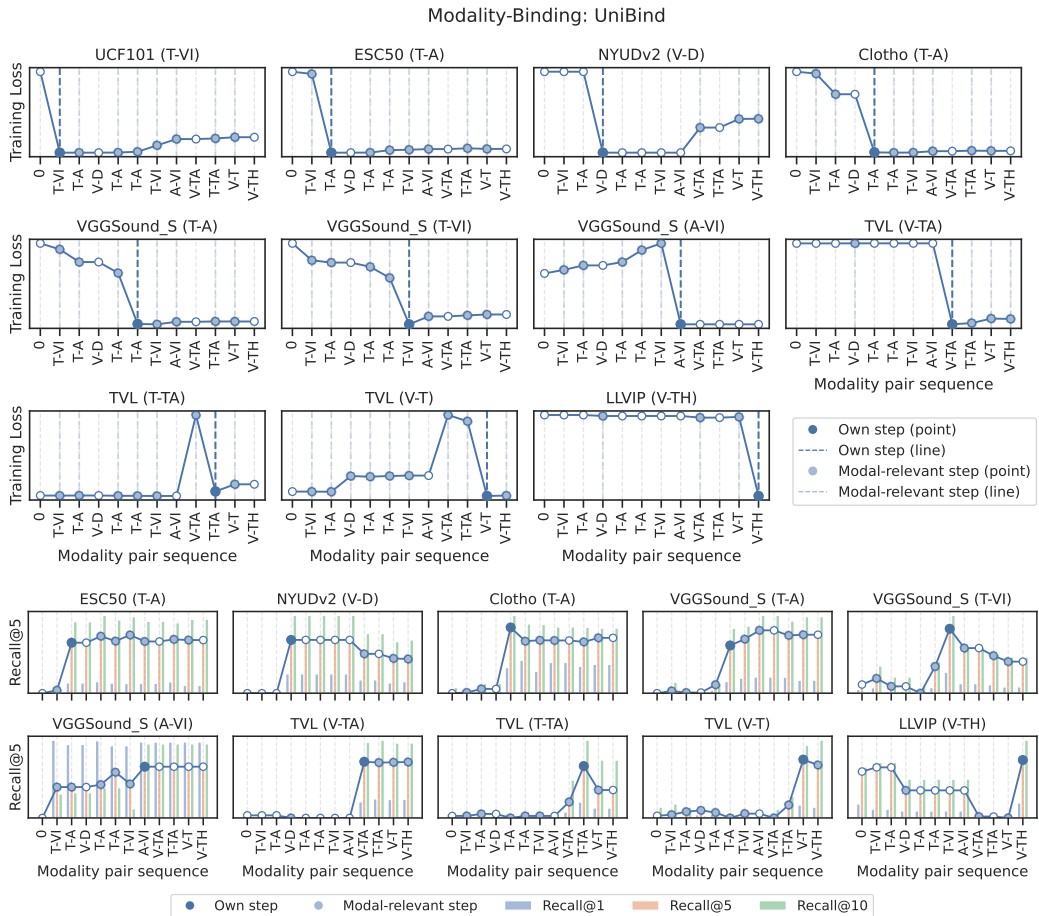

Figure 12: Detailed experimental results of the training loss (*the top three rows*) and recall (*the last two rows*) over a sequence of modality pair data, with *UniBind* as the modality-binding backbone. Notably, the y-values are normalized for better visualization.

