# OpenReview forum: "Continual Multimodal Contrastive Learning"
_NeurIPS.cc/2025/Conference — NeurIPS 2025 poster_

### Official Review · Reviewer_yJSf · 2025-06-23

**Clarity:** 3
**Significance:** 3
**Originality:** 4
**Rating:** 5
**Confidence:** 4

**Summary:**

This paper introduces a novel framework for Continual Multimodal Contrastive Learning (CMCL), where modality-paired data arrives sequentially and integrated without catastrophic forgetting. To address this challenge, the authors propose Dual-sided Null Space (DNS) projection, which modifies gradient updates to lie in null spaces derived from previous modalities’ representations. The authors mainly discussed the stability (retaining knowledge) and plasticity (adapting to new modality information) of CMCL. Theoretically, the authors derive two upper bounds characterizing the loss in stability and plasticity, and empirically validate DNS on extensive multimodal datasets using three modality-binding backbones, achieving strong performance across different tasks with minimal training overhead.

**Questions:**

Most of the concerns have been raised in Strength and Weakness sections. It would be great if the authors could specifically address the following questions:

1. Can DNS be extended to settings where 3+ modalities are introduced jointly at a step, rather than only pairs?
2. Have you considered using explicit factorizations to separate shared and exclusive subspaces to account for the modality gap?
3. What is the assumption on the encoders? How would a poorly pretrained encoders effect the representation quality throughout the time as the update depends on previous steps?
4. Could the authors provide some visualization analysis of how the representation space evolves over training steps?
5. How robust is the proposed method under scenarios where the modality pairs could be severely imbalanced or noisy? Can the authors briefly discuss this scenario?

Overall, I enjoyed reading the paper and look forward to authors’ responses.

**Ethical Concerns:**

["NO or VERY MINOR ethics concerns only"]

**Final Justification:**

I appreciate the authors’ response and I’m keeping my current positive score.

**Limitations:**

yes

**Quality:**

4

**Strengths And Weaknesses:**

This paper addresses an interesting and important problem of continual learning for multimodal contrastive models beyond the standard bimodal setups. The proposed method (DNS) is theoretically justified and emprically effective, showing improved stability and plasticity with notable gaps compared to existing continuous learning frameworks. Its ability to plug into different modality-binding backbones and operate in a replay-free manner enhances its practicality and generaliazbility. The contributions are promising and the findings in the paper can lead to interesting discussions and extensions for multimodal community. The pros and cons are discussed below:

**Pros**

- **Strong motivation and interesting problem**: The paper makes a valuable conceptual contribution by formally defining CMCL and distinguishing it from traditional continual learning and multimodal learning. The clear definitions of stability and plasticity is interesting in  evaluating and understanding the setting of continual multimodal learning.
- **Theoretical justification:** The derivation of dual-sided gradient projection is carefully grounded in linear algebraic principles, leveraging null space projectors to preserve past knowledge.
- **Strong evaluation:** The experiments span 7 diverse datasets across vision, audio, text, thermal, depth, and tactile modalities. DNS consistently outperforms continual learning baselines across different tasks. The proposed method demonstrates superior performance in terms of information retainment and adaptation.
- **Flexibility with existing frameworks:** The proposed method is shown to work with ImageBind, LanguageBind, and UniBind, indicating its compatibility with a wide class of pretrained modality-binding architectures. This makes the method flexible in integration with many existing or future works.
- **Efficiency:** The authors have also demonstrated the efficiency of the proposed method with minimum overhead compared to the vanilla method, in contrast to the replay based methods.

**Cons**

- **Generalization beyond 2-modality pairs**: Although the method supports sequential training on modality pairs, the extension to scenarios where more than two modalities are jointly introduced (have a sequence of three or more modalities) is not addressed. This could have additional bottlenecks to more complex CMCL settings (e.g., sensor systems) where we could obtain >2 continuous modalities at the same time.
- **Pretrained encoders**: The method assumes modality encoders can be reused or extended without explicitly retraining from scratch. It’s unclear whether the modality-specific encoders (e.g., gm3) are pretrained or jointly trained. It is unclear how does the effectiveness of the pretrained encoders impact the performance.
- **Modality gaps or factorized latent spaces**: A key challenge in multimodal learning is modality gap, i.e., the representational divergence between modalities. The proposed method operates under the assumption of a unified embedding space, but the subspace relationships (shared vs. exclusive latent dimensions) are not examined. A discussion or experiment here would be great.
- **Visualizations**: The paper lacks visualizations (e.g., t-SNE of modality representations across steps, heatmaps of alignment matrices) that would help diagnose how the method enhances or preserves the representation space during the updates.
- **Imbalanced or noisy multimodal scenario**: There’s limited analysis or discussion on when or why DNS might fail, such as in highly imbalanced modality data scenarios, noisy data, or misaligned priors.

I enjoyed reading the paper overall. I have summarized some questions in the later section.

---

> ### Author Rebuttal · Authors · 2025-07-31
>
> Thanks for your professional and careful review. We respond to your concerns or questions as follows.
>
> > W1: Generalization beyond 2-modality pairs: Although...same time.
>
> **Response**
>
> Thanks for the insightful comment! Our method potentially can extend to >2 modalities at one training step. The core alignment is defined as the inner product of two modalities, yet can be extended for more modalities. Recent methods, like VAST [1], employ the onmi-modality, which aggregates all modalities, to align with an anchor modality (e.g., text) for pair-wise contrastive learning. DNS inherently can be extended to these methods. GRAM [2], a seminar work, proposes to align more than two modalities simultaneously by minimizing the volume of a Gram matrix $\mathbf{G} = \mathbf{Z}^\top \mathbf{Z}$. Every entry in this matrix is exactly the inner product of two modalities, aligning with the definition of our alignment score in Equation 5. The difference is that, one modality (e.g., $m_1$) should be aligned with other modalities together. Therefore, the alignment can be represented as:
>
> $(\mathbf{Z}^{m_1})^\top \mathbf{Z}^{\perp m_1} = (\mathbf{Z}^{m_1}_{\ast})^\top (\mathbf{W}^{m_1})^\top\mathbf{W}^c \mathbf{Z}^c\_{*}$,
>
> $\mathbf{W}^c =[\mathbf{W}^{m_2}, \mathbf{W}^{m_3}, \ldots, \mathbf{W}^{m_{|\mathcal{M}|}}]$,
>
> $\mathbf{Z}^c\_{\ast} = \mathrm{diag}(\mathbf{Z}^{m_2}\_{\ast}, \mathbf{Z}^{m_3}\_{\ast}, \ldots, \mathbf{Z}^{m_{|\mathcal{M}|}}\_{\ast})$.
>
> $\mathbf{Z}^{m_1} \in \mathbb{R}^{d \times n}$ represents modality $m_1$, and $\mathbf{Z}^{\perp m_1} \in \mathbb{R}^{d \times (|\mathcal{M}|-1)n}$ includes features of all other modalities. With this formulation, conceptually, we can replace the modal projector $\tilde{\mathbf{P}}$ with the one constructed from $\mathbf{Z}^c_*$, aggregated data from all other modalities, while maintaining the self-modal projector $\mathbf{P}'$. We hope that this possible solution is helpful and can address your concerns.
>
> [1] Chen, et al. "Vast: A vision-audio-subtitle-text omni-modality foundation model and dataset." NeurIPS 2023.
> [2] Cicchetti, et al. "Gramian multimodal representation learning and alignment." ICLR 2025.
>
> > W2: Pretrained encoders: The...performance.
>
> **Response**
>
> Thanks! Using a pretrained encoder is a practical implementation for our method, leveraging well-developed open-source model weights. In our implementation, modality-specific encoders are jointly trained, thus possessing certain alignment. However, our analysis does not assume this, allowing for separately trained encoders. We evaluate our method on different types of encoders, like ImageBind and LanguageBind. We can observe that the base model (the pretrained model) with better joint training would lead to a better performance (here LanguageBind beats ImageBind at most cases). Therefore, the pretrained encoders can affect the performance. A better pretrained encoder would lead to better performance. Otherwise, if we use the isolated trained encoders, the performance would also decrease.
>
> > W3: Modality gaps or factorized latent spaces: A...would be great.
>
> **Response**
>
> Thanks! The modality gap phenomenon can be preserved by contrastive learning [3]. Within a unified space, different modalities might lie in different regions [4], i.e., the modality gap. The unified space indicates that we can use a simple inner product to estimate the alignment. Our research starts from and focuses on multimodal alignment scores, using a conventional contrastive learning objective. Therefore, according to [3], our method can also preserve the modality gap. We also visualize the representations and  conduct some experiments. We compute the modality centroid by averaging different modality representations, and estimate their similarity. We have the following table:
> |Centroid similarity|Vanilla|CILA|DNS|
> |-|-|-|-|
> |t=0|9.35|9.35|9.35|
> |t=T|4.14|6.67|9.09|
>
> We can observe that the modality gap is preserved or even worse for vanilla and cila methods (lower similarity). However, DNS exhibits a slight similarity drop. We will add this analysis in the revised version of the paper.
>
> [3] Liang, et al. "Mind the gap: Understanding the modality gap in multi-modal contrastive representation learning." NeurIPS 2022.
> [4] Schrodi, et al. "Two Effects, One Trigger: On the Modality Gap, Object Bias, and Information Imbalance in Contrastive Vision-Language Models." ICLR 2025.
>
> > W4: Visualizations: The...during the updates.
>
> **Response**
>
> Thanks!  Because most of the data are pairwise, we do not depict the heatmaps.  We visualize the representations with t-SNE across steps to observe their distribution changes as suggested. We are willing to share the insights with you: the modality gap is still preserved while is slightly changing when continually learning. This is reasonable and aligns with our experiments on W3 and its response. DNS does not influence the learning objective, so it will preserve the modality gap from contrastive learning [3]. However, DNS showcases better robustness, alleviating the enlarging modality gap of other methods. We will add this analysis in the revised version.
>
> > W5: Imbalanced or noisy multimodal scenario: There...priors
>
> **Response**
>
> Thanks! It is interesting to test our method under imbalanced, noisy, or misaligned multimodal scenarios. We conduct experiments to evaluate performance across varying levels of noise and misalignment, providing a comprehensive analysis of DNS’s robustness.
>
> **Imbalanced Modality Data Scenarios**: We construct a specialized dataset with modality pairs of text-video (t0) and text-audio (t1, t2, t3), to assess performance in imbalanced scenarios. We test text-video at step 1 (less overlap) and text-audio at step 2 (most overlap), where greater overlap indicates worse forgetting and larger performance drops. As shown in the table below, DNS consistently outperforms Vanilla and CILA, demonstrating superior robustness:
>
> |Imbalanced|Best (at it own step)|Vanilla|CILA|DNS|
> |-|-|-|-|-|
> |t0(T-VI)|87.72|85.34|85.54|87.92|
> |t1(T-A)|49.25|41.00|43.75|47.50|
>
> **Misaligned Pairs**: We create misaligned modality pairs by exchanging features at ratios of 0.01, 0.05, 0.1, 0.2, 0.3, 0.4, and 0.5. As misalignment increases, performance decreases across methods. However, DNS consistently outperforms CILA, demonstrating stronger robustness:
>
> |Misalign|acc(CILA)|acc(DNS)|BWT_acc(CILA)|BWT_acc(DNS)|Recall(CILA)|Recall(DNS)|BWT_R(CILA)|BWT_R(DNS)|
> |-|-|-|-|-|-|-|-|-|
> |0.01|49.39|52.04|-3.52|0.11|38.03|40.32|-1.72|-1.35|
> |0.05|46.45|46.98|-1.81|2.93|37.46|39.41|-1.32|-1.22|
> |0.1|41.46|43.54|0.36|2.00|36.21|38.50|-0.9|-0.90|
> |0.2|33.33|35.18|1.26|1.39|33.91|36.63|-1.49|-1.85|
> |0.3|28.73|29.27|-0.8|1.46|32.12|34.09|-1.94|-2.32|
> |0.4|19.91|24.97|-5.59|0.17|29.43|32.84|-2.6|-2.64|
> |0.5|17.32|19.78|-6.48|-4.30|27.58|30.55|-3.86|-4.03|
>
> **Noisy Data**: We add Gaussian noise to the training dataset at scales of 0.01, 0.05, 0.1, 0.2, 0.3, 0.4, and 0.5. Performance decreases with increasing noise, but DNS outperforms CILA across all metrics, showcasing greater robustness:
>
> |Noise|acc(CILA)|acc(DNS)|BWT_A(CILA)|BWT_A(DNS)|Recall(CILA)|Recall(DNS)|BWT_R(CILA)|BWT_R(DNS)|
> |-|-|-|-|-|-|-|-|-|
> |0.01|50.37|52.18|-3.12|-0.40|38.46|40.65|-1.25|-1.11|
> |0.05|47.42|50.74|-5.31|-1.79|34.59|39.79|-4.41|-2.66|
> |0.1|43.89|50.44|-7.94|-0.69|34.19|40.21|-4.07|-0.66|
> |0.2|39.6|43.82|-9.54|-6.32|32.58|37.29|-5.09|-3.56|
> |0.3|34.45|39.48|-13.8|-11.15|30.62|35.77|-7.41|-4.30|
> |0.4|31.92|39.22|-14.36|-11.63|29.3|34.89|-8.53|-5.26|
> |0.5|28.52|37.31|-18.62|-11.29|28.7|33.35|-7.94|-6.72|
>
> These results confirm that while performance degrades in extreme scenarios for all methods, DNS consistently demonstrates greater robustness and effectiveness.
>
> > Q1: Can DNS be extended to settings where 3+ modalities are introduced jointly at a step, rather than only pairs?
>
> **Response**
>
> Yes. DNS is capable of handling more than 2 modalities if we extract the gradients and conduct the projection as explained in W1 and the corresponding response.
>
> > Q2: Have you considered using explicit factorizations to separate shared and exclusive subspaces to account for the modality gap?
>
> **Response**
>
> This is an interesting question! We did not consider explicitly factoring the spaces. We attribute the explicit factorizations to the learning objectives, while our method mainly focuses on how to project the gradient to keep the alignment scores. These alignment scores can be maintained or manipulated by different learning strategies. We also conduct experiment for clarification, as shown in the response to W3. A possible way for explicit factorizations can be implemented in the hyperbolic space [5] that preserves the hierarchies.
>
> [5] Ramasinghe, et al. "Accept the modality gap: An exploration in the hyperbolic space." CVPR 2024.
>
> > Q3: What is the assumption on the encoders? How would a poorly pretrained encoders effect the representation quality throughout the time as the update depends on previous steps?
>
> **Response**
>
> Generally, we have no assumption about the encoders. However, a better pretrained encoder indeed leads to a better performance. We detail this in W2 and the corresponding response.
>
> > Q4: Could the authors provide some visualization analysis of how the representation space evolves over training steps?
>
> **Response**
>
> Sure! Due to the policy, we cannot send you the image. We share the insights with you in W4 and the corresponding response.
>
> > Q5: How robust is the proposed method under scenarios where the modality pairs could be severely imbalanced or noisy? Can the authors briefly discuss this scenario?
>
> **Response**
>
> Yes! From our experiments, DNS is more robust to imbalanced/noisy/misaligned scenarios. A more detailed analysis can be found in W5 and the corresponding response.
>
> We authors sincerely thank you for your professional review attitude and comments. If you have other problems, we are happy to address them to polish this work.

---

> > ### Comment · Reviewer_yJSf · 2025-08-04
> > **Rebuttal response**
> >
> > Thank you for the comprehensive response and the additional results provided. I appreciate the thoroughness of your analysis. Good luck with your rebuttal.

---

> ### Author Response · Authors · 2025-08-04
> **Response to Reviewer yJSf**
>
> We appreciate your professional review! Your feedback is very valuable and will be reflected in our final version!

---

### Official Review · Reviewer_2wop · 2025-07-01

**Clarity:** 4
**Significance:** 3
**Originality:** 3
**Rating:** 4
**Confidence:** 3

**Summary:**

This paper introduces and formulates the problem of Continual Multimodal Contrastive Learning (CMCL), where models must learn from a sequence of datasets with different modality pairs without forgetting previously learned knowledge. To address this, the authors define two principles for CMCL: stability (retaining old knowledge) and plasticity (learning new knowledge). The primary contribution is a novel, theoretically-grounded method called Dual-sided Null Space (DNS) projection. DNS works by projecting the parameter gradients onto a null space where they are prevented from interfering with previously learned modality alignments. The method is evaluated on seven datasets using three different modality-binding models as backbones. The results demonstrate superior performance over several continual learning baselines in both stability and plasticity.

**Questions:**

1. The construction of null space projectors relies on accumulating feature covariance matrices from past steps. As the number of continual learning steps increases, could this accumulation process lead to information loss or degradation, thereby affecting the precision of the projectors over long learning sequences?

2. The visualization in Figure 2 is difficult to interpret, as the distinction between the top and bottom plots is unclear. Could you please revise the figure or add a more detailed explanation to clarify its key message?

3. Could you provide a more detailed analysis of your method's memory complexity? Specifically, how does the storage cost for the feature covariance matrices scale with the number of tasks and feature dimensions, and how does this compare to the memory requirements of a typical replay-based method?

**Ethical Concerns:**

["NO or VERY MINOR ethics concerns only"]

**Final Justification:**

My concerns, especially regarding the method's novelty and its explanation, have been addressed. I maintain my positive review.

**Limitations:**

Yes

**Paper Formatting Concerns:**

The paper is well-formatted with no significant issues.

**Quality:**

3

**Strengths And Weaknesses:**

Strengths

1.	The paper clearly defines the Continual Multimodal Contrastive Learning (CMCL) problem and proposes a theoretically-grounded solution (DNS) to address catastrophic forgetting.

2.	The proposed method is rigorously derived and supported by theoretical proofs for stability and plasticity.

3.	The method is extensively evaluated on seven datasets against multiple baselines, showing strong performance.

Weaknesses

1.	While the application to CMCL is novel, the core idea of using null space projection on gradients for continual learning has been explored before. The main novelty lies in the sophisticated "dual-sided" adaptation for the multimodal setting, but the underlying principle is not entirely new.

2. The DNS method avoids the memory buffer of replay-based methods but requires storing and updating feature covariance matrices to construct the projectors. The paper does not provide a detailed analysis of how the memory footprint of these matrices scales with a very large number of continual learning steps and high feature dimensions.

3. The authors rightly point out in their limitations (Appendix I) that the paper does not examine the generalization errors of the proposed method in downstream tasks. It is plausible that strictly constraining gradient updates to preserve past alignments might limit the model's ability to learn more generalizable features that could be beneficial for an unseen downstream task.

---

> ### Author Rebuttal · Authors · 2025-07-31
>
> Thanks for your professional and careful review. We respond to your concerns or questions as follows.
>
> **Response**
>
> > W1: While the application to CMCL is novel, the core idea of using null space projection on gradients for continual learning has been explored before. The main novelty lies in the sophisticated "dual-sided" adaptation for the multimodal setting, but the underlying principle is not entirely new.
>
> Thank you for recognizing our contribution to CMCL and for the opportunity to clarify our method compared to conventional null space projection! At a high level, our method shares some conceptual similarities with null space projection. However, the conventional null space ($\mathbf{AX=0}$) is insufficient for CMCL ($\mathbf{AXB=0}$), as CMCL requires alignment across multimodalities from dual sides ($\mathbf{A}$ and $\mathbf{B}$). The conventional null space only cares `one side`, i.e., one modality ($\mathbf{A}$).  Deriving a rigorous solution for CMCL and implementing it effectively are highly non-trivial, far beyond a straightforward adaptation of null space projection, also reflected in our detailed theoretical derivation. That indicates the significance of our "dual-sided" projection, which breaks the constraints of null space in theory and advances the practical implementation of CMCL. We call our method `dual-sided null space` (DNS) to highlight its distinct contribution, rather than framing it as a mere adaptation. In fact, we point out this distinction from null space in Line 177 and will further clarify it in the revised paper.
>
> > W2: The DNS method avoids the memory buffer of replay-based methods but requires storing and updating feature covariance matrices to construct the projectors. The paper does not provide a detailed analysis of how the memory footprint of these matrices scales with a very large number of continual learning steps and high feature dimensions.
>
> **Response**
>
> Thanks! In Section 3.3 ("Extension to Any Steps"), we explain that our method maintains a global feature covariance matrix, updated solely with data from the current learning step, without storing historical data. This update process is independent of the number of steps, ensuring high efficiency and preventing computational cost accumulation as learning progresses. Regarding feature dimensions, the covariance matrix is defined in $\mathbb{R}^{d \times d}$, where $d$ is the fixed and typically small feature dimension compared to the dataset size. Once training begins, $d$ remains constant. Thus, with a fixed feature dimension and step-independent updates, the computational cost remains low and stable throughout training.
>
> > W3: The authors rightly point out in their limitations (Appendix I) that the paper does not examine the generalization errors of the proposed method in downstream tasks. It is plausible that strictly constraining gradient updates to preserve past alignments might limit the model's ability to learn more generalizable features that could be beneficial for an unseen downstream task.
>
> **Response**
>
> Thanks for the comment! We understand your concern. In fact, in the research community of machine/deep learning theory, rigorously examining the generalization errors/bounds about contrastive learning is very challenging [1]. Recent studies often focus on bimodal scenarios (e.g., image and text) and linearized contrastive loss [2, 3], whereas the widely adopted non-linear contrastive loss still remains intractable in the current research and beyond our current scope. Actually, we empower DNS with certain flexibility if you want to learning the new more. The projector can be adjusted via $\lambda_{min}$. A large $\lambda_{min}$ will provide more plasticity for learning more generalizable features. We also provide a comprehensive analysis on hyperparameters in Appendix F. Hope these could address your concern.
>
> [1] HaoChen, Jeff Z., et al. "Provable guarantees for self-supervised deep learning with spectral contrastive loss." NeurIPS, 2021.
> [2] Zhang, Qi, et al. "On the generalization of multi-modal contrastive learning." ICML, 2023.
> [3] Xue, Yihao, et al. "Understanding the robustness of multi-modal contrastive learning to distribution shift." ICLR, 2024.
>
> > Q1: The construction of null space projectors relies on accumulating feature covariance matrices from past steps. As the number of continual learning steps increases, could this accumulation process lead to information loss or degradation, thereby affecting the precision of the projectors over long learning sequences?
>
> **Response**
>
> No, our method is efficient and independent of the number of training steps. The update of feature covariance matrices is equivalent to computing directly on data from all past steps. Therefore, there is also no information loss in our matrix update. We clarify this in Section 3.3 ("Extension to Any Steps") with a proof in Apendix B.5.
>
> > Q2: The visualization in Figure 2 is difficult to interpret, as the distinction between the top and bottom plots is unclear. Could you please revise the figure or add a more detailed explanation to clarify its key message?
>
> **Response**
>
> Sure! We are willing to improve Figure 2 for better clarity and understanding of our method. Due to image update policies, we describe the revisions in text. Figure 2 depicts the model parameter update process for vanilla training versus DNS training. At step $t$, the gradient $\nabla \mathbf{W}\_t$ is computed from modality pair data (e.g., image and audio). In vanilla training, this gradient directly updates the parameters, overwriting parts overlapping with the previously learned gradient $\nabla \mathbf{W}_{t-1}$, particularly around the image modality, which leads to forgetting in continual learning. In contrast, DNS avoids this overlap by projecting the new gradient $\nabla \mathbf{W}_t$ onto the non-overlapping subspace, preserving prior knowledge while incorporating the new. We will further clarify these distinctions in the revised paper.
>
> > Q3: Could you provide a more detailed analysis of your method's memory complexity? Specifically, how does the storage cost for the feature covariance matrices scale with the number of tasks and feature dimensions, and how does this compare to the memory requirements of a typical replay-based method?
>
> **Response**
>
> Thanks for your interest! We are willing to explain this (also refer to W2). The memory required for feature covariance matrices is fixed at the start of training, depending only on the number of modalities ($|\mathcal{M}|$) and the feature dimension ($d$), which is significantly smaller than the dataset size ($B$). Thus, our method’s memory cost can be represented as $\mathcal{O}(|\mathcal{M}|\cdot d^2)$. In contrast, typical replay-based methods have memory costs of $\mathcal{O}(|\mathcal{M}|\cdot B \cdot d)$, that scale with data size, growing substantially as data increases.
>
> We authors sincerely thank you for your professional review attitude and comments. If you have other questions, we are happy to address them to polish this work.

---

> > ### Comment · Reviewer_2wop · 2025-08-04
> > **Rebuttal Comment by Reviewer 2wop**
> >
> > Thank you for your detailed response. You have effectively addressed my concerns, and I will maintain my positive review.

---

> ### Author Response · Authors · 2025-08-04
> **Response to Reviewer 2wop**
>
> Thank you for your constructive comments! They definitely help us polish this work!

---

### Official Review · Reviewer_gDQw · 2025-07-02

**Clarity:** 3
**Significance:** 3
**Originality:** 3
**Rating:** 4
**Confidence:** 3

**Summary:**

The paper proposes a novel approach to Continual Multimodal Contrastive Learning (CMCL), a problem at the intersection of multimodal learning and continual learning. In CMCL, a model is trained incrementally on emerging multimodal data, where modality pairs (such as vision-text, vision-audio, and text-audio) are introduced sequentially. The core challenge addressed is the trade-off between stability (retaining knowledge from previously learned modalities) and plasticity (learning new modality pairs without forgetting previous ones). The authors introduce a new optimization method that projects parameter gradients onto subspaces to maintain both stability and plasticity during incremental learning. They also provide theoretical guarantees and two bounds that address these aspects. The empirical evaluation of their method across multiple datasets shows superior performance compared to other continual learning methods.

**Questions:**

Please check the above section.

**Ethical Concerns:**

["NO or VERY MINOR ethics concerns only"]

**Final Justification:**

After the rebuttal, the authors solved most of my concerns, and I will keep my positive score.

**Limitations:**

Please check the above section.

**Paper Formatting Concerns:**

no formatting concerns

**Quality:**

3

**Strengths And Weaknesses:**

Strength:

1.  The topic of continual multimodal contrastive learning is meaningful and with great practical value.  The proposed method is grounded in real-world issues such as the cost of training models from scratch with large multimodal datasets and the need to incorporate emerging data.
2. The authors provide a detailed theoretical analysis, with clear definitions of stability and plasticity, and detailed proofs to support their gradient-projection method.
3. The authors conduct extensive experiments on seven datasets and compare their approach with existing baselines, showing strong empirical results.

Weakness:

1. The proposed gradient-projection method is complex, and while the theory is well-supported, it remains unclear how easy it would be to implement and integrate into existing models or systems. It would be to provide discussion and experiments about the training efficiency, like FLOPs.
2. It would be more convincing to provide hyper-parameter sensitivity analysis for \lambda_{min}.

---

> ### Author Rebuttal · Authors · 2025-07-31
>
> Thanks for your professional and careful review. We respond to your concerns or questions as follows.
>
> > W1: The proposed gradient-projection method is complex, and while the theory is well-supported, it remains unclear how easy it would be to implement and integrate into existing models or systems. It would be to provide discussion and experiments about the training efficiency, like FLOPs.
>
> **Response**
>
> Thanks! We provide implementation details, including algorithm flow and pseudocode, in Appendix A. Notably, it is easy to integrate into existing models with minor code (only two lines of code for gradient projection). To confirm this, we also adapt our method for three existing encoders, i.e., ImageBind, LanguageBind, and UniBind, in our evaluation. We also provide the efficiency analysis in Section 4.3, indicating its efficiency with negligible time cost. As you suggested, we also provide FLOPs analysis.
> ||Complete|Additional |
> |-|-|-|
> |FLOPs|23.73 T|268.44 M|
>
> As can be seen, by using ImageBind as the backbone with a batch size of 64, the complete procedure requires 23.73T FLOPs (one forward & backward), while feature matrix maintenance and projection consume only 268.44M FLOPs, confirming our method’s efficiency. We are happy to include further evidence as needed.
>
> > W2: It would be more convincing to provide hyper-parameter sensitivity analysis for $\lambda_{min}$.
>
> **Response**
>
> Yes, we agree with you and we have already included the hyper-parameter analysis in Appendix F, examining various $\lambda_{\min}$ as suggested. If you have further concerns or questions, kindly let us know. We can provide clarifications.
>
> We sincerely appreciate your professional and constructive feedback. Your insightful comments have greatly helped improve our manuscript. Should you have any further questions or suggestions, we would be more than happy to address them to further refine and enhance this work.

---

> > ### Comment · Reviewer_gDQw · 2025-08-05
> >
> > Thanks for the authors' response. My concerns are mostly addressed. I will keep my positive score.

---

> > > ### Author Response · Authors · 2025-08-05
> > > **Response to Reviewer gDQw**
> > >
> > > Thank you for your positive attitude and thoughtful comments, which have significantly helped improve our paper!

---

### Official Review · Reviewer_82Az · 2025-07-03

**Clarity:** 3
**Significance:** 3
**Originality:** 3
**Rating:** 4
**Confidence:** 2

**Summary:**

This paper defines Continual Multimodal Contrastive Learning, a new problem of learning from a stream of multimodal data pairs. The authors propose a novel gradient projection method, Dual-sided Null Space, to prevent catastrophic forgetting by ensuring new updates do not interfere with prior knowledge. The method is supported by strong theoretical guarantees and empirical results showing it significantly outperforms continual learning baselines in balancing stability and plasticity.

**Questions:**

As the number of continual learning steps (t) increases, how does the memory and computational cost required to store and compute the cumulative feature covariance matrix (Z<t) for the projectors scale?
The paper defines stability as maintaining the alignment score for old data pairs. While this is a clear metric, is it sufficient to guarantee that the overall semantic structure of the embedding space is preserved?  For example, will it affect downstream tasks?

**Ethical Concerns:**

["NO or VERY MINOR ethics concerns only"]

**Final Justification:**

The authors’ response has to some extent addressed my concerns, but no additional experimental data were provided to substantiate it; therefore I am maintaining my score.

**Limitations:**

Although the DNS method is more efficient than replay-based approaches, it requires constructing and maintaining a feature covariance matrix from all prior learning steps and computing projections via SVD approximation. Meanwhile, the paper mainly focus on retrieval and classification tasks and gains marginal improvement. Furthermore, the method's generalization and effectiveness on more complex downstream tasks, such as multimodal reasoning, have not been validated.

**Quality:**

3

**Strengths And Weaknesses:**

It introduces and formally defines Continual Multimodal Contrastive Learning (CMCL), establishing a new and important research direction.  It proposes a novel method, DNS, that is both theoretically justified and empirically proven to be effective at learning continually without forgetting.

However, the method itself relies on learning from pairs of data, which isn't really how people learn—we take in everything at once. So the big question is whether this pair-by-pair approach can ever truly grasp the complex connections you only get when combining three or more senses[1].  Therefore, not updating existing knowledge does not seem to be a  long-term strategy.

[1] Cicchetti, Giordano, et al. "Gramian Multimodal Representation Learning and Alignment." arXiv preprint arXiv:2412.11959 (2024).

---

> ### Author Rebuttal · Authors · 2025-07-31
>
> Thanks for your professional and careful review. We respond to your concerns or questions as follows.
>
> > W1: However, the method itself relies on learning from pairs of data, which isn't really how people learn—we take in everything at once. So the big question is whether this pair-by-pair approach can ever truly grasp the complex connections you only get when combining three or more senses[1]. Therefore, not updating existing knowledge does not seem to be a long-term strategy.
>
> **Response**
>
> Thanks for your insightful feedback! Preserving alignment across modality pairs is challenging due to the complex interactions of modal gradients, as shown in Equation 5. Extending this to multiple modalities within a single training step further complicates theoretical and practical guarantees. However, our method, DNS, potentially handles alignment beyond pairwise data. Below is our analysis.
>
> There are two primary methods to align multiple modalities simultaneously. The first method aggregates features from all modalities (e.g., via concatenation and projection) into an omni-modality feature, aligned with an anchor modality (e.g., text) [1]. In this case, DNS can be easily adapted by substituting one modality with the omni-modality feature. The second method, exemplified by GRAM [2], directly aligns multiple modalities using a Gram matrix, where each entry represents the inner product of two modalities, aligning with our alignment score definition. DNS can be conceptually adapted to GRAM as follows.
> Given the Gram matrix $\mathbf{G} = \mathbf{Z}^\top \mathbf{Z}$, where $\mathbf{Z}$ is the concatenated multimodal features, GRAM optimizes the parameters of one modality ($m_1$) by aligning it with other modalities ($m_2, m_3, \ldots$). The alignment score is defined as $\mathbf{Z}^{m_1\top} \mathbf{Z}^{\perp m_1} \in \mathbb{R}^{n \times (|\mathcal{M}|-1)n}$, where $\mathbf{Z}^{m_1} \in \mathbb{R}^{d \times n}$ represents modality $m_1$, and $\mathbf{Z}^{\perp m_1} \in \mathbb{R}^{d \times (|\mathcal{M}|-1)n}$ includes features of all other modalities. This can be expressed as:
>
> $(\mathbf{Z}^{m_1})^\top \mathbf{Z}^{\perp m_1} = (\mathbf{Z}^{m_1}_{\ast})^\top (\mathbf{W}^{m_1})^\top\mathbf{W}^c \mathbf{Z}^c\_{*}$,
>
>  $\mathbf{W}^c =[\mathbf{W}^{m_2}, \mathbf{W}^{m_3}, \ldots, \mathbf{W}^{m_{|\mathcal{M}|}}]$,
>
> $\mathbf{Z}^c\_{\ast} = \mathrm{diag}(\mathbf{Z}^{m_2}\_{\ast}, \mathbf{Z}^{m_3}\_{\ast}, \ldots, \mathbf{Z}^{m_{|\mathcal{M}|}}\_{\ast})$.
>
> Using this formulation, DNS can adapt by replacing  the modal projector $\tilde{\mathbf{P}}$ with a new one built upon $\mathbf{Z}^c_*$, aggregated data from all other modalities, while maintaining the self-modal projector $\mathbf{P}'$. We hope that this possible solution is helpful and can address your concerns.
> This capability demonstrates that DNS can scale to multiple modalities, supporting its long-term viability. We will expand on this in Section 3.3 to clarify our method’s broad applicability.
>
> [1] Chen, Sihan, et al. "Vast: A vision-audio-subtitle-text omni-modality foundation model and dataset." NeurIPS, 2023.
> [2] Cicchetti, Giordano, et al. "Gramian multimodal representation learning and alignment." ICLR, 2025.
>
> > Q1: As the number of continual learning steps (t) increases, how does the memory and computational cost required to store and compute the cumulative feature covariance matrix (Z<t) for the projectors scale?
>
> **Response**
>
> Thank you for your thorough review and interest! Our method does not incur accumulating costs when updating the feature covariance matrix across learning steps. As detailed in Section 3.3, "Extension for Any Steps", we update the feature covariance matrix ($\mathbf{Z}_{<t}$) at each step using only the current features, without needing to store or recompute past features. This matrix, in $\mathbb{R}^{d \times d}$, maintains a fixed size, ensuring no increase in storage requirements. Therefore, our approach remains computationally efficient with minimal overhead. We will add this explanation in the final version of the paper.
>
> > Q2: The paper defines stability as maintaining the alignment score for old data pairs. While this is a clear metric, is it sufficient to guarantee that the overall semantic structure of the embedding space is preserved? For example, will it affect downstream tasks?
>
> **Response**
>
> Thank you for your thoughtful question! The semantic structure in multimodal alignment is not explicitly defined but is implicitly captured through alignment scores, as seen in models like CLIP and ImageBind, consistent with CMCL. From this perspective, our method can preserve the semantic structure of previous modality data because our goal is to keep the previous alignment scores.  However, we also emphasize the learning plasticity. The embedding space is refined with continual learning; a rough semantic structure also becomes more fine-grained along with the training. For example, a factor initially representing "dog" may be refined to distinguish "chihuahua" after further training, with additional factors like "poodle" emerging as the null space is updated. Therefore, such refinement impacts downstream tasks, likely improving performance when tasks depend on precise multimodal alignment.
>
> > L1: Although the DNS method is more efficient than replay-based approaches, it requires constructing and maintaining a feature covariance matrix from all prior learning steps and computing projections via SVD approximation. Meanwhile, the paper mainly focuses on retrieval and classification tasks and gains marginal improvement. Furthermore, the method's generalization and effectiveness on more complex downstream tasks, such as multimodal reasoning, have not been validated.
>
> **Response**
>
> Thanks! Regarding your first concern, we clarify the efficiency of updating the feature covariance matrix, which is independent of the number of learning steps (see Q1 and its related response). The efficient analyses (Section 4.3) show that maintaining projectors incurs minimal computational cost. In terms of performance, our method surpasses baselines by up to 3.98% in Acc, and 3.18% in BWT. These improvements are also noted by other reviewers (Reviewer gDQw, 2wop, and yJSf). For more complex downstream tasks, we agree that more tasks would further strengthen our method. However, adapting refined representations for advanced models (e.g., MLLMs) is resource-intensive, and due to time constraints, we could not include more results. We also clarify this in Q2 that enhanced alignment is expected to improve performance in downstream tasks like multimodal understanding that rely on precise alignment. Your suggested investigation will be reflected in the final version.
>
> We authors sincerely thank you for your professional review attitude and comments. If you have other questions, we are happy to address them to polish this work.

---

> > ### Comment · Reviewer_82Az · 2025-08-05
> >
> > Thanks for the detailed response. Looking forward to seeing more discussion on the method and additional experiments in the paper.

---

> > > ### Author Response · Authors · 2025-08-05
> > > **Response to Reviewer 82Az**
> > >
> > > Thank you for your thoughtful comments. We are pleased to see that your concerns have been addressed. We will include these revisions in our paper.

---

### Author Response · Authors · 2025-08-01
**General Response**

We appreciate the reviewers’ insightful comments and constructive feedback on our manuscript. We are pleased to receive positive ratings from all the reviewers. Furthermore, we are delighted to learn that the reviewers found the research problem to be significant (Reviewers 82Az, gDQw, 2wop, and yJSf), the core idea to be interesting and flexible (Reviewers 82Az, gDQw, 2wop, and yJSf), the theoretical analysis to be solid (Reviewers 82Az, gDQw, 2wop, and yJSf), and the experiments to be convincing (Reviewers 82Az, gDQw, 2wop, and yJSf). Based on the reviews, we provide a general response to the points raised by multiple reviewers and individual responses below to address each reviewer’s concerns.

(1) Regarding the questions about the experiments, we have taken the following actions:

- For Reviewer yJSf, we investigate the impact of pretrained encoders, interesting questions about imbalanced/noisy/misaligned scenarios and modality gaps, which are comprehensively supported with additional experiments.

- For Reviewers gDQw and 2wop, we demonstrate the efficiency of our method with additional efficiency evaluation (FLOPs) and memory analysis.

- For Reviewer gDQw, we clarify the analysis of hyperparameters provided in the Appendix.


(2) We have addressed the questions about the idea and technical details as follows:

- For Reviewers 82Az and yJSf, we clarify the extension of the proposed method to the scenario of more than two modalities, providing two possible adaptations.

- For Reviewers 82Az, gDQw, and 2wop, we clarify the efficiency and losslessness of updating the feature covariance matrices.

- For Reviewers 82Az and 2wop, we clarify the generalizability of the proposed method.

- For Reviewers 2wop and yJSf, we illustrate Figure 2 more clearly for a better understanding of our method and representation visualization.

We sincerely thank all the reviewers for their constructive suggestions. Please feel free to let us know if further details/explanations would be helpful.

Yours truly,
Authors of #969

---

### Note · Authors · 2025-08-15

Dear PC, SAC, AC, and Reviewers,

We sincerely thank all reviewers for their constructive feedback, which has greatly polished our manuscript. We are encouraged by the overall positive evaluation and thoughtful comments from all reviewers, and appreciate the valuable suggestions that helped us clarify and improve key aspects of our work. This broad agreement highlights the significance and robustness of our approach.

The core contribution of our work lies in introducing a significant research problem in multimodal contrastive learning: rather than training models on all data in a single process or from scratch, we focus on continually learning from sequential modality pairs, enabling efficient and incremental optimization while addressing catastrophic forgetting. With rigorous theoretical analysis, we explicitly balance stability (retaining knowledge from prior modality alignments) and plasticity (adapting to new pairs) through a novel dual-sided gradient projection method. We clarify that our method operates directly on parameter gradients without requiring data replay, making it efficient and applicable to diverse modality-binding backbones like ImageBind, LanguageBind, and UniBind. At the same time, our method can potentially extend to handle multiple modalities (>2) in a single training step, paving the way for broader applications and motivating future research in the multimodal learning community.

In closing, we sincerely appreciate the invaluable suggestions from all reviewers, which have substantially improved the scientific rigor and clarity of our manuscript. We will carefully incorporate their comments into the final version if accepted, and believe our work offers a novel and effective paradigm for continual multimodal contrastive learning.

Sincerely,
The Authors

---

### Decision · Program_Chairs · 2025-09-17

**Decision:**

Accept (poster)

**Comment:**

This paper introduces Continual Multimodal Contrastive Learning (CMCL), a new problem setting where multimodal contrastive models are optimized over sequentially arriving modality pairs rather than trained from scratch. To address this, the authors propose a theoretically grounded gradient projection method that balances stability and plasticity.

This paper received overall positive reviews from the reviewers, with all final scores in the borderline accept to accept range. The authors made a sincere effort during the rebuttal to address key concerns, and reviewers generally acknowledged the responses as satisfactory. No major objections remained after the discussion phase.

While additional experimental validation could further strengthen the work, the current version presents a meaningful contribution. Overall, the AC finds the paper suitable for acceptance based on the positive consensus and the effective resolution of reviewer concerns, though some points may benefit from refinement in the final revision.